# Teacher Guided Training: An Efficient Framework for Knowledge Transfer

**Manzil Zaheer[†], Ankit Singh Rawat[†], Seungyeon Kim,**
**Chong You, Himanshu Jain, Andreas Veit, Rob Fergus, Sanjiv Kumar**
Google Research and DeepMind, New York, USA
{manzilzaheer,ankitsrawat,seungyeonk}@google.com
{cyou,himj,aveit,robfergus,sanjivk}@google.com

## Abstract

The remarkable performance gains realized by large pretrained models, e.g., GPT-3, hinge on the massive amounts of data they are exposed to during training. Analogously, distilling such large models to compact models for efficient deployment also necessitates a large amount of (labeled or unlabeled) training data. In this paper, we propose the *teacher-guided training* (TGT) framework for training a high-quality compact model that leverages the knowledge acquired by pretrained *generative* models, while obviating the need to go through a large volume of data. TGT exploits the fact that the teacher has acquired a good representation of the underlying data domain, which typically corresponds to a much lower dimensional manifold than the input space. Furthermore, we can use the teacher to explore input space more efficiently through sampling or gradient-based methods; thus, making TGT especially attractive for limited data or long-tail settings. We formally capture this benefit of proposed data-domain exploration in our generalization bounds. We find that TGT can improve accuracy on several image classification benchmarks as well as a range of text classification and retrieval tasks.

## 1 Introduction

Recent general purpose machine learning models (e.g., BERT (Devlin et al., 2019), DALL-E (Ramesh et al., 2021), SimCLR (Chen et al., 2020a), Perceiver (Jaegle et al., 2021), GPT-3 (Brown et al., 2020)), trained on broad data at scale, have demonstrated adaptability to a diverse range of downstream tasks. Despite being trained in unsupervised (or so-called self-supervised) fashion, these models have been shown to capture highly specialized information in their internal representations such as relations between entities Heinzerling & Inui (2021) or object hierarchies from images (Weng et al., 2021).

Despite their impressive performance, the prohibitively high inference cost of such large models prevents their widespread deployment. A standard approach to reducing the inference cost while preserving performance is to train a compact (student) model via knowledge distillation (Bucilua et al., 2006; Hinton et al., 2015) from a large (teacher) model. However, existing distillation methods require a large amount of training data (labeled or unlabeled) for knowledge transfer. For each data point, the teacher must be evaluated, making the process computationally expensive (Xie et al., 2020d; He et al., 2021; Sanh et al., 2019a). This is compounded by the need to repeat the distillation process separately for every downstream task, each with its own training set. Enabling efficient distillation is thus an important challenge. Additionally, minimizing the number of distillation samples would especially benefit low-data downstream tasks, e.g., those with long-tails.

Another inefficiency with standard distillation approaches is that within each evaluation of the teacher, only the final layer output (aka logits) is utilized. This ignores potentially useful internal representations which can also be levered for knowledge transfer. Various extensions have been proposed in the literature along these lines (see, e.g., (Sun et al., 2020; Aguilar et al., 2020; Li et al., 2019; Sun et al., 2019) and references therein). However, despite their success, such extensions mostly use the teacher model in a black-box manner, and do not fully utilize the domain understanding it contains (Cho & Hariharan, 2019; Stanton et al., 2021). In these approaches, the teacher is used *passively* as the input sample distribution remains fixed and does not adapt to the student

---

[†]Equal contribution.

model performance. Consequently, these forms of distillation do not lead to faster training of a high-performance student model.

In this work, we go beyond the passive application of large teacher models for training compact student models, and leverage the domain understanding captured by the teacher to generate new informative training instances that can help the compact model achieve higher accuracy with fewer samples and thus enable reduced training time. In particular, we propose the *teacher guided training* (TGT) framework for a more efficient transfer of knowledge from large models to a compact model. TGT relies on the fact that teacher's internal representation of data often lies in a much smaller dimensional manifold than the input dimension. Furthermore, we can use teacher to help guide training by identifying the directions where the student's current decision boundary starts to diverge from that of the teacher, e.g., via backpropagating through the teacher to identify regions of disagreement.

We provide a theoretical justification for the TGT algorithm, showing that leveraging the data representation of large models ensures better generalization for the student. Given $n$ instances in a $D$-dimensional space the generalization gap for learning a Lipschitz decision boundary of a classification task decays only as $\mathcal{O}\big(n^{-\frac{1}{D}}\big)$ (Györfi et al., 2002). In contrast, provided that the large model learns a good data representation in a $d$-dimensional latent space, the TGT framework realizes a generalization gap of $\mathcal{O}\big(n^{-\frac{1}{d}} + \mathcal{W}(\mathcal{D}, \mathcal{D}^t)\big)$, where $\mathcal{W}(\mathcal{D}, \mathcal{D}^t)$ denotes the Wasserstein distance between the data distribution $\mathcal{D}$ and the distribution $\mathcal{D}^t$ learned by the generative teacher model. Typically $d \ll D$, thus TGT ensures much faster convergence whenever we use a high-quality generative teacher; making TGT especially attractive for low-data or long-tail regimes.

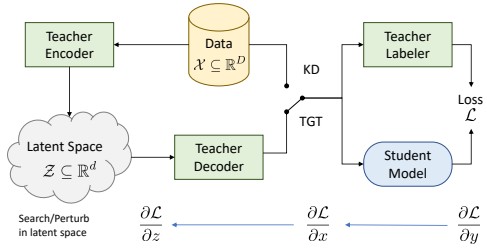

Figure 1: An overview of the proposed *teacher guided training* (TGT) framework. Given a learning task, the framework leverages a large teacher with a pretrained **generator** and **labeler** that exhibits high performance on the task. In particular, we assume that the generator consists of an **encoder** and a **decoder**. TGT performs three key operations during student model training: (1) Given an original training instance, by using the teacher generator, identify a novel task-relevant instance. We search for informative instances in the lower dimensional latent space, where we can propagate the gradient to. (2) Obtain (soft) labels for the original and newly generated training instances from the teacher labeler; and (3) Minimize the student training objective that depends on the original and newly generated instances along with their labels produced by the teacher labeler.

To realize TGT, we take advantage of the fact that most of the unsupervised pretrained models like Transformers, VAE, and GANs have two components: (1) an *encoder* that maps data to a latent representation, and (2) a *decoder* that transforms the latent representation back to the original data space. We utilize this latent space for the data representations learned by the teacher model to efficiently search for the regions of mismatch between the teacher and student's decision boundaries. This search can take the form of either (i) a zero-order approach involving random perturbation or (ii) a first-order method exploring along the direction of the gradient of a suitably defined distance measure between the teacher and student models.

Many pretrained models, particularly in NLP such as T5 (Raffel et al., 2020), can also provide labels for a downstream task and act as a sole teacher. However, our approach is sufficiently general to utilize separate pretrained models for generative and discriminative (labeler) functions (cf. Fig. 1), e.g., we employ a BiGAN as generator and an EfficientNet as labeler for an image classification task.

Our main contributions are summarized as follows:

1. We introduce the TGT framework, a conceptually simple and scalable approach to distilling knowledge from a large teacher into a smaller student. TGT adaptively changes the distribution of distillation examples, yielding higher performing student models with fewer training examples.

2. We provide theoretical justifications for utilizing the latent space of the teacher generator in the TGT framework, which yields tighter generalization bounds.

3. We empirically show the superiority of TGT to existing state-of-the-art distillation methods on both vision and NLP tasks, unlike most prior work which is specialized to one domain.

## 2 RELATED WORK

Our proposed TGT framework can be considered a form of data augmentation where data is dynamically added at points of current discrepancy between the teacher and student. Next, we provide a brief overview of how data augmentation has been used in the context of distillation and distinguish our work from these existing efforts.

**Using pseudo labels.** The earliest line of work involves using *consistency regularization* (Sajjadi et al., 2016; Tarvainen & Valpola, 2017) to obtain pseudo labels for unlabelled data while a model is expected to make consistent predictions on an unlabeled instance and its augmented versions, cf. (Miyato et al., 2019; Xie et al., 2020a; Sohn et al., 2020; Zhu et al., 2021, inter alia). Another approach is *self-training* (Xie et al., 2020d; Du et al., 2021) where one learn a *smaller* teacher model on the labeled data which then generates pseudo labels for a large relevant unlabeled set. A large student model is then trained on both labeled and pseudo labeled sets. *Label propagation* (Iscen et al., 2019) is another direction where unlabeled instances receive pseudo labels based on neighboring labeled instances in a suitably constructed similarity graph.

Furthermore, prior work on *learning to teach* (Fan et al., 2018; Raghu et al., 2021; Pham et al., 2021), dynamically updates the teacher so as to provided more valuable pseudo labels based on the student loss. Such an interactive approach presents a challenging optimization problem and potentially opens up the door for borrowing techniques from reinforcement learning. In contrast, our work focuses on the setting where high-quality pretrained teacher model is fixed throughout the training. We focus on a setting where updating the large teacher model is prohibitively costly or undesirable as such a model would potentially be used to distill many student models. Moreover, many large models like GPT-3 may only be available through API access, thus making it infeasible to update the teacher.

**Using pretrained models.** One can use large pretrained class conditional generative models like BigGAN (Brock et al., 2019) or VQ-VAE2 (Razavi et al., 2019) to generate more data for augmentation. Despite evidence (Webster et al., 2019) that GANs are not memorizing training data, using them to simply augment the training dataset has limited utility when training ResNets (Ravuri & Vinyals, 2019b;a). Lack of diversity (Arora et al., 2017) in data generated by GANs, especially among high density regions (Arora et al., 2018), is a potential reason for this. In contrast, we use generative models to adaptively explore the local region of disagreement between teacher and student as opposed to blindly sampling from the generative model. This way we circumvent the excessive reliance on samples from high density regions which often have low diversity.

Another line of work by Chen et al. (2020b) combines unsupervised/self-supervised pretraining (on unlabeled data) with SimCLR-based approach (Chen et al., 2020a), task-specific finetuning (on labeled data), and distillation (natural loss on labeled and distillation loss on unlabeled data). Our work is very close to this line of work with two key differences: (1) We assume access to a very high-quality teacher, which is potentially trained on a much larger labeled set, to provide pseudo labels; (2) We go beyond utilizing a given relevant unlabeled dataset and explore the *dynamic generation* of domain-specific unlabeled data by leveraging the representations learned by pretrained models. Additionally, we develop a theoretical framework to establish the utility of unlabeled data instances for student training, specifically the instances generated based on teacher learned representations.

**Using both pseudo labels and pretrained models.** The GAL framework (He et al., 2021) previously considered generating training instances by using pretrained generator models along with pseudo-labelers. However, the GAL framework generates these new instances in an offline manner at the beginning of student training. In contrast, our approach (cf. Fig. 1) generates new informative instances in an online fashion to attain high-performance and reduce training time for the student.

Recently, MATE-KD (Rashid et al., 2021) also used a generator model to obtain new training instances based on the student's current performance (by looking at the divergence between the student and teacher predictions). However, there are two key differences between our TGT approach and the MATE-KD framework: First, their method updates the teacher so as to find adversarial examples for the students, which can cause the generator to drift away from the true data distribution. Second, they introduce perturbations in the input space itself and do not leverage the latent space of the teacher, which is the crux of our method. See Appendix A for further details.

Notably KDGAN (Wang et al., 2018) leverages a GAN during distillation. However, it samples examples from a GAN without taking student's performance into account. Heo et al. (2019); Dong et al. (2020) search for adversarial examples during distillation. However, their search also does

not depend on student's performance, resulting in wasteful exploration of those regions of the input spaces where the student is already good. Further, unlike TGT, they search examples in the input space which is often inefficient due to the large ambient dimension of the input space.

Finally, data-free KD approaches (Nayak et al., 2019; Yoo et al., 2019; Chen et al., 2019) use only synthetically generated data for knowledge distillation. Unlike TGT, in such approaches, the synthetic data distribution is updated at each epoch, which causes the student model to lose the information over epochs and experience accuracy degradation (Binici et al., 2022). In this framework, Micaelli & Storkey (2019) targeted generating samples that would cause maximum information gain to the student when learned, however, it also suffers from similar drawbacks as MATE-KD noted above.

## 3 TEACHER GUIDED TRAINING

We begin by formally introducing our setup in Section 3.1. We then describe our proposed TGT framework in Section 3.2 and present its theoretical analysis in Section 3.3.

### 3.1 PROBLEM SETUP

In this paper, we focus on a multiclass classification task where given an instance $x \in \mathcal{X}$ the objective is to predict its true label $y \in \mathcal{Y} := [K]$ out of $K$ potential classes. Let $\mathcal{D} := \mathcal{D}_{X,Y}$ denote the underlying (joint) data distribution over the instance and label spaces for the task. Moreover, we use $\mathcal{D}_X$ and $\mathcal{D}_{Y|X=x}$ to denote the marginal distribution over the instance space $\mathcal{X}$ and the conditional label distribution for a given instance $x$, respectively. A classification model $f : \mathcal{X} \to \mathbb{R}^K$, with $f(x) = (f(x)_1, \ldots, f(x)_K)$, takes in an input instance $x$ and yields scores for each of the $K$ classes. Finally, we are given a (tractable) loss function $\ell : \mathbb{R}^K \times [K] \to \mathbb{R}$ which closely approximates model's misclassification error on an example $(x, y)$, e.g., softmax-based cross-entropy loss.

Given $n$ i.i.d. labeled samples $\mathcal{S}_n^{\text{labeled}} := \{(x_i, y_i)\}_{i \in [n]}$ generated from $\mathcal{D}$ and a collection of allowable models $\mathcal{F}$, one typically learns a model via *empirical risk minimization* (ERM):

$$\widehat{f}_n = \arg\min_{f \in \mathcal{F}} \frac{1}{n} \sum\nolimits_{i \in [n]} \ell(f(x_i), y_i). \tag{1}$$

In our TGT setup, we further assume access to a high quality *teacher model*, which has:

- **Teacher generator.** A *generative* component that captures $\mathcal{D}_X$ well, e.g., a transformer, VAE, or ALI-GAN. This usually consists of an encoder $\mathsf{Enc} : \mathcal{X} \to \mathbb{R}^d$ and a decoder $\mathsf{Dec} : \mathbb{R}^d \to \mathcal{X}$.

- **Teacher labeler.** A *classification network*, denoted by $h : \mathcal{X} \to \mathbb{R}^K$, with good performance on the underlying classification task. In general, our framework allows for $h$ to be either a head on top of the teacher generator or an independent large teacher classification model.

Given $\mathcal{S}_n^{\text{labeled}}$ and such a teacher model, our objective is to learn a high-quality compact *student* (classification) model in $\mathcal{F}$, as assessed by its misclassification error on $\mathcal{D}$.

### 3.2 PROPOSED APPROACH

To train a student model $f \in \mathcal{F}$, we propose to minimize:

$$R_f^{\text{TGT}}(\mathcal{S}_n^{\text{labeled}}) := \frac{1}{n} \sum_{i \in [n]} \left( \ell(f(x_i), y_i) + \ell_{\text{d}}(f(x_i), h(x_i)) \right) + \frac{1}{m} \sum_{j \in [m]} \ell_{\text{d}}(f(\tilde{x}_j), h(\tilde{x}_j)) \tag{2}$$

where $\ell_{\text{d}} : \mathbb{R}^K \times \mathbb{R}^K \to \mathbb{R}$ is a loss function that captures the mismatch between two models $f$ and $h$, and $\tilde{\mathcal{S}}_m = \{\tilde{x}_j\}_{j \in [m]}$ is introduced in subsequent passage. The first term, $\ell(f(x_i), y_i)$, corresponds to standard ERM problem (cf. Eq. (1)). The subsequent terms, $\ell_{\text{d}}(f(x_i), h(x_i))$ and $\ell_{\text{d}}(f(\tilde{x}_j), h(\tilde{x}_j))$, do not make use of labels. In particular, the second term, $\ell_{\text{d}}(f(x_i), h(x_i))$, corresponds to the standard knowledge distillation where the teacher $h$ provides supervision for the student $f$.

We introduce a novel third term, $\ell_{\text{d}}(f(\tilde{x}_j), h(\tilde{x}_j))$, where $\tilde{\mathcal{S}}_m = \{\tilde{x}_j\}$ is generated based on $\mathcal{S}_n = \{x_i\}$. Here, we want to generate informative instances $\tilde{\mathcal{S}}_m$ that will help student learn faster, e.g., points *on* the data manifold where the student disagrees with the teacher. In other words, we want to find $\tilde{x}$ as follows:

$$\tilde{x} \in \arg\max_{x \in \mathcal{X}} \ell_{\text{d}}(f(x), h(x)) \text{ such that } p_{\mathcal{D}_X}(x) \geq \lambda \tag{3}$$

Note that the objective and constraint in Eq. (3) ensure that we select an instance where the student and teacher disagree and the instance belongs to a region where true data distribution assigns a non-trivial mass, respectively. Based on this, we propose two specific approaches to generate the novel samples $\tilde{\mathcal{S}}_m$:

1. **Isotropically perturb in latent space:**

$$\tilde{x} = \mathsf{Dec}(\mathsf{Enc}(x) + \nu) \quad \text{where } \nu \sim \mathcal{N}(0, \sigma^2 \mathbb{I}_d). \tag{4}$$

   This can be regarded as a zero-order search in the latent space, which satisfies the constraint of remaining within the data manifold.

2. **Gradient-based exploration:** Run a few iterations of gradient ascent on Eq. (3) in order to find the example that diverges most with teacher. To enforce the constraint, we run the gradient ascent in the latent space of the teacher generator as opposed to performing gradient ascent in the instance space $\mathcal{X}$, which might move the perturbed point out of the data manifold. For a high-quality teacher generator, the latent space should capture the data manifold well. To implement this we need to backprop all the way through the student and teacher-labeler to the teacher-decoder, as shown in Fig. 1. Mathematically, it involves the following three operations:

$$z := \mathsf{Enc}(x); \qquad z \;\leftarrow z + \eta \nabla_z \ell_{\mathrm{d}}\left(f(\mathsf{Dec}(z)), h(\mathsf{Dec}(z))\right); \qquad \tilde{x} := \mathsf{Dec}(z). \tag{5}$$

   This is akin to a first-order search in the latent space.

**Extension to discrete data.** Note that perturbing an instance from a discrete domain, e.g., text data, is not as straightforward as in a continuous space. Typically, one has to resort to expensive combinatorial search or crude approximations to perform such perturbations (Tan et al., 2020; Zang et al., 2020; Ren et al., 2019). Interestingly, our approach in Eq. (4) provides a simple alternative where one performs the perturbation in the latent space which is continuous. On the other hand, in gradient based exploration, we assume that $\mathcal{X}$ is a differentiable space in order to calculate necessary quantities such as $\frac{\partial f(x)}{\partial x}$ in Eq. (5). This assumption holds for various data such as images and point clouds but not for discrete data like text. We can, however, circumvent this limitation by implementing weight sharing between the output softmax layer of the teacher's decoder Dec and the input embedding layer of the student $f$ (and also to teacher labeler $h$ when an independent model is used). Now, one can bypass discrete space during the backward pass, similar to ideas behind VQ-VAE (Hafner et al., 2019). Note that, during forward pass, we still need the discrete representation for decoding, e.g., using beam search.

Finally, we address the superficial resemblance between our approach and adversarial training. For latter, the goal is to learn a robust classifier, i.e., to increase margin. Towards this, for any $x$, one encourages model agreement in its local neighborhood $B_r(x)$, i.e., $f(x') = f(x), \forall x' \in B_r(x)$. One needs to carefully choose small enough neighborhood by restricting $r$, so as to not cross the decision boundary. In contrast, we are not looking for such max-margin training which has its own issues (Nowak-Vila et al., 2021). We simply desire *global* agreement between the teacher and student, i.e., $f(x') = h(x')$, $\forall x'$. As a result, we can explore much bigger regions as long as we remain on the data manifold, i.e., $p_{\mathcal{D}_X}(x)$ is non-trivially large.

### 3.3 VALUE OF GENERATING SAMPLES VIA THE LATENT SPACE

Now, we formally show how leveraging the latent space can help learning. For this exposition, we assume $\mathcal{X} = \mathbb{R}^D$. Furthermore, for directly learning in the input space, we assume that our function class $\mathcal{F}$ corresponds to all Lipschitz functions that map $\mathbb{R}^D$ to $\mathbb{R}^K$. For any such function $f \in \mathcal{F}$, existing generalization bounds take the form (Devroye et al., 2013; Mohri et al., 2018):

$$R_{\ell,f}(\mathcal{D}) \le R_{\ell,f}(\mathcal{S}_n) + \underbrace{\mathfrak{R}_n(\mathcal{G}_{\ell,\mathcal{F}})}_{\le \mathcal{O}(n^{-1/D})} + \mathcal{O}\left(\sqrt{\log(1/\delta)/n}\right),$$

where $R_{\ell,f}(\mathcal{D})$ is true population risk of the classifier, $R_{\ell,f}(\mathcal{S}_n)$ is empirical risk, and $\mathfrak{R}_n(\mathcal{G}_{\ell,\mathcal{F}})$ is the Rademacher complexity of the induced function class $\mathcal{G}_{\ell,\mathcal{F}}$, which is known in our case to be $\mathcal{O}(n^{-1/D})$ (see Appendix B for more details). Note that any reduction in the Rademacher term would imply a smaller generalizing gap, which is our goal.

In our TGT framework, we assume availability of a teacher that is able to learn a good representation for the underlying data distribution. In particular, we assume that, for $x \in \text{supp}(\mathcal{D}_X)$, we have

$$\|\text{Dec} \circ \text{Enc}(x) - x\| \leq \epsilon, \tag{6}$$

i.e., for $x$, applying the decoder $\text{Dec}$ on the latent representation of $x$, as produced by the encoder $\text{Enc}$, leads to a point $\text{Dec} \circ \text{Enc}(x) \in \mathcal{X}$ that approximates $x$ with a small error.

This ability of teacher generator to model the data distribution using latent representation can be used to reduce the complexity of the function class needed. Specifically, in TGT framework, we leverage the teacher decoder to restrict the function class to be a composition of the decoder function $\text{Dec}$ and a learnable Lipschitz function operating on the latent space $\mathbb{R}^d$. Since $d \ll D$, this leads to a function class with much lower complexity. Next, we formally capture this idea for distillation with both the original samples $\mathcal{S}_n$ sampled from $\mathcal{D}_X$ as well as the novel samples $\tilde{\mathcal{S}}$ introduced by the teacher generator. In what follows, we only consider the distillation losses and ignore the first loss term (which depends on true labels). Our analysis can be easily extended to take the latter term into account (e.g., by using tools from Foster et al. (2019)).

We start with the standard distillation in the following result. See Appendix C.1 for the details.

**Theorem 3.1.** *Suppose a generative model with* $\text{Enc}$ *and* $\text{Dec}$ *satisfies the approximation guarantee in Eq. (6) for* $\mathcal{D}_X$. *Let* $\text{Dec}$ *and teacher labeler* $h$ *be Lipschtiz functions, and the distillation loss* $\ell_{\text{d}}$ *satisfies Assumption C.1. Then, with probability at least* $1 - \delta$, *the following holds for any* $f \in \mathcal{F}$.

$$R_{\ell,f}(\mathcal{D}) \leq R_{\ell_{\text{d}},f}^h(\mathcal{S}_n) + \underbrace{\mathfrak{R}_n(\mathcal{G}_{\ell_{\text{d}},\mathcal{F}}^{h,\text{Dec}})}_{\leq \mathcal{O}(n^{-1/d})} + \mathcal{O}\left(\frac{\sqrt{\log(1/\delta)}}{\sqrt{n}}\right) + L\epsilon + \mathcal{O}\left(\sqrt{K}\mathbb{E}_{\mathcal{D}_X}\left[\|\mathcal{D}_{Y|X} - h(X)\|_2\right]\right).$$

*where* $L$ *is the effective Lipschitz constant of* $\mathcal{G}_{\ell_{\text{d}},\mathcal{F}}^{h,\text{Dec}} = \{z \mapsto \ell_{\text{d}}(f \circ \text{Dec}(z), h \circ \text{Dec}(z)) : f \in \mathcal{F}\}$ — *an induced function class which maps* $\mathbb{R}^d$ *(latent space of generator) to* $\mathbb{R}$.

Thus, we can reduce the Rademacher term from $O(n^{-1/D})$ to $O(n^{-1/d})$, which yields a significant reduction in sample complexity. However, as the teacher model is not perfect, a penalty is incurred in terms of reconstruction error $L\epsilon$ and prediction error $\mathcal{O}\left(\sqrt{K}\mathbb{E}_{\mathcal{D}_X}\left[\|\mathcal{D}_{Y|X} - h(X)\|_2\right]\right)$.

Thus far, we have not leveraged the fact that we can also use the teacher to generate additional samples. Accounting for using samples $\tilde{\mathcal{S}}_n$ (cf. Section 3.2), one can obtain similar generalization gap for the distillation based on the teacher generated samples:

**Theorem 3.2.** *Let* $\tilde{\mathcal{S}}_n = \{\tilde{x}_i\}_{i \in [n]}$ *be* $n$ *i.i.d. samples generated by the the TGT framework, whose distribution be denoted by* $\tilde{\mathcal{D}}_X$. *Further, let* $\tilde{f}_n \in \mathcal{F}$ *denote the student model learned via distillation on* $\tilde{\mathcal{S}}_n$, *with* $h$ *as the teacher model and* $\ell_{\text{d}}$ *be the distillation loss satisfying Assumption C.1. Then, with probability at least* $1 - \delta$, *we have*

$$R_{\ell,f}(\mathcal{D}) \leq R_{\ell_{\text{d}},\tilde{f}_n}^h(\tilde{\mathcal{S}}_n) + \underbrace{\tilde{\mathfrak{R}}_n(\mathcal{G}_{\ell_{\text{d}},\mathcal{F}}^{h,\text{Dec}})}_{\leq \mathcal{O}(n^{-1/d})} + \mathcal{O}\left(\sqrt{\frac{\log(1/\delta)}{n}}\right) + \mathcal{W}(\mathcal{D}_X, \tilde{\mathcal{D}}_X)$$

$$+ \mathcal{O}\left(\sqrt{K}\mathbb{E}_{D_X}\left[\|D_{Y|X} - h(X)\|_2\right]\right), \quad \text{where } \mathcal{G}_{\ell_{\text{d}},\mathcal{F}}^{h,\text{Dec}} \text{ is defined in Thm. 3.1}$$

Please see Appendix C.2 for a more precise statement and proof of Thm. 3.2. Comparing with the generalization gap for standard distillation (cf. Thm. 3.1), the generalization gap for TGT in Thm. 3.2 does not have the reconstruction error related term $L\epsilon$. Thus, by working with the samples with exact latent representation, TGT avoids this reconstruction error penalty. On the other hand, generalization gap for TGT does have an additional term $\mathcal{W}(D_X, \tilde{D}_X)$, capturing the mismatch between the original data distribution and the distribution of the samples used by TGT.

As a guiding principle, Thm. 3.2 dictates that one should select a teacher generator that minimizes $\mathcal{W}(\mathcal{D}_X, \tilde{\mathcal{D}}_X)$. Similarly, the teacher labeler should ensure small prediction error $\mathcal{O}\left(\sqrt{K}\mathbb{E}_{D_X}\left[\|D_{Y|X} - h(X)\|_2\right]\right)$ for the underlying classification task.

**Motivation for gradient-based exploration.** Our theoretical results so far do not throw light on the particular utility of the gradient-based exploration in Eq. (5). In this regard, we provide variance-based

| | Approach | Architecture | Balanced Accuracy | # parameters | FLOPs |
|---|---|---|---|---|---|
| **ImageNet K-LT** | Logit adjustment loss* (Menon et al., 2021b) | ResNet-50 | 50.4 | 26 M | 4.1 B |
| | LDAM-DRS-RSG (Wang et al., 2021) | ResNeXt-50 | 51.8 | 25 M | 4.2 B |
| | DRAGON + Bal'Loss (Samuel et al., 2021) | ResNet-10 | 46.5 | 5.4 M | 819 M |
| | DRAGON + Bal'Loss (Samuel et al., 2021) | ResNet-50 | 53.5 | 26 M | 4.1 B |
| | *Teacher (labeler) model* | EfficientNet-b3 | 79.2 | 12 M | 1.8 B |
| | One-hot | MobileNetV3-0.75 | 35.5 | 4.01 M | 156 M |
| | Distillation | MobileNetV3-0.75 | 47.2 | 4.01 M | 156 M |
| | TGT (random) | MobileNetV3-0.75 | 53.2 | 4.01 M | 156 M |
| | TGT (gradient-based) | MobileNetV3-0.75 | 53.3 | 4.01 M | 156 M |
| **SUN397-LT** | LWS (Kang et al., 2020) | ResNeXt-50 | 33.9 | 25 M | 4.2 B |
| | DRAGON + Bal'Loss (Samuel et al., 2021) | ResNet-101 | 36.1 | 42 M | 7.6 B |
| | *Teacher (labeler) model* | EfficientNet-b3 | 65.3 | 12 M | 1.8 B |
| | One-hot | MobileNetV3-0.75 | 39.3 | 4.01 M | 156 M |
| | Distillation | MobileNetV3-0.75 | 42.2 | 4.01 M | 156 M |
| | TGT (random) | MobileNetV3-0.75 | 44.3 | 4.01 M | 156 M |
| | TGT (gradient-based) | MobileNetV3-0.75 | 44.7 | 4.01 M | 156 M |

Table 1: Performance of TGT and various baselines on long-tail image classification benchmarks (see Appendix E for results on Places-LT). Rows with * denote results taken from Menon et al. (2021b) and the rest were taken from Samuel et al. (2021). We report top-1 accuracy on balanced eval sets. We also state the number of model parameters and inference cost (in terms of FLOPs) for all the methods. Note that TGT leads to performance improvements over standard distillation on all three datasets, particularly for ImageNet-LT where the teacher generator models the task distribution well. TGT also often outperforms stated baselines that rely on much larger and expensive models.

generalization bounds (Maurer & Pontil, 2009) in Appendix C.3. Such bounds suggest that, besides minimizing the discrepancy $\mathcal{W}(\mathcal{D}_X, \tilde{\mathcal{D}}_X)$, an ideal $\tilde{\mathcal{D}}_X$ should reduce the variance of $\ell_{\mathrm{d}}\big(f(\tilde{x}), h(\tilde{x})\big)$ for newly generated instances. Incidentally, the sampling approach realized by the gradient-based exploration in Eq. (5) aims to achieve this: it controls for $\mathcal{W}(\mathcal{D}_X, \tilde{\mathcal{D}}_X)$ by operating in the latent space of a good quality teacher generative model and minimizes variance by finding instances with high loss values through gradient ascent, thereby striking a desired balance between the two objectives. See Appendix C.3 for a detailed discussion.

# 4 EXPERIMENTS

We now conduct a comprehensive empirical study of our TGT framework in order to establish that TGT (i) leads to high accuracy in transferring knowledge in low data/long-tail regimes (Section 4.1); (ii) effectively increases sample size (Section 4.2); and (iii) has wide adaptability even to discrete data domains such as text classification (Section 4.3) and retrieval (Section 4.4).

## 4.1 LONG-TAIL IMAGE CLASSIFICATION

**Setup.** We evaluate TGT by training student models on three benchmark long-tail image classification datasets: ImageNet-LT (Liu et al., 2019c), SUN-LT (Patterson & Hays, 2012), Places-LT (Liu et al., 2019c) We employ off-the-shelf teacher models, in particular BigBiGAN (ResNet-50) (Donahue & Simonyan, 2019) and EfficientNet-B3 (Xie et al., 2020c) as the teacher generator and teacher labeler models, respectively. We utilize MobileNetV3 (Howard et al., 2019) as compact student model architecture. The teacher-labeler model is self-trained on JFT-300M (Sun et al., 2017), and then finetuned on the task-specific long-tail dataset. The teacher generator is trained on the unlabelled full version of ImageNet (Russakovsky et al., 2015).

**Results.** The results[1] are reported in Table 1 compared with similar sized baselines (we ignored gigantic transformer models). We see that TGT is able to effectively transfer knowledge acquired by the teacher during its training with the huge amount of data into a significantly smaller student model, which also has lower inference cost. TGT considerably improves the performance across the board over standard distillation, even on Sun-LT and Places-LT whose data distribution *does not* exactly match to the distribution that the teacher's generator was trained with. That said, the gains from TGT are more pronounced when the mismatch between the task data distribution and the

---

[1] Results for Places-LT and additional baselines for ImageNet-LT and SUN-LT are in Appendix E.

| Method | Architecture | Amazon-5 | | IMDB | MNLI | Yelp-5 | |
|---|---|---|---|---|---|---|---|
| | | 2.5k | 3M | | | 2.5k | 650k |
| UDA (Random Init) (Xie et al., 2020b) | BERT base | 55.8 | - | - | - | 58.6 | - |
| UDA (Pretrained) (Xie et al., 2020b) | BERT base | 62.9 | - | - | - | 67.9 | - |
| Layer-wise Distillation (Sun et al., 2020) | MobileBERT | - | - | 93.6 | 83.3 | - | - |
| MATE-KD (Rashid et al., 2021) | DistilBERT | - | - | - | 85.8 | - | - |
| *Teacher (labeler) model* | RoBERTa large | - | 67.6 | 96.2 | 90.6 | - | 72.0 |
| One-hot (Random Init) | DistilBERT | 44.3 | 53.6 | 50.0 | 63.0 | 50.4 | 58.1 |
| One-hot (Pretrained) | DistilBERT | 55.9 | 66.3 | 93.6 | 84.1 | 59.1 | 67.3 |
| Distillation (Random Init) | DistilBERT | 56.5 | 65.3 | 87.9 | 77.4 | 54.8 | 69.5 |
| Distillation (Pretrained) | DistilBERT | 60.2 | 66.8 | 94.0 | 84.5 | 63.2 | 71.4 |
| TGT (Random Init) | DistilBERT | 61.3 | 66.6 | 91.0 | 79.3 | 62.0 | 70.4 |
| TGT (Pretrained) | DistilBERT | **64.6** | **67.1** | **95.4** | **86.0** | **68.6** | **71.7** |

Table 2: Performance of TGT and various baselines from the literature on four text classification benchmarks. For student model training, we show results for task-specific finetuning on both randomly initialized and pretrained DistilBERT models. Note that TGT (Pretrained) — TGT with a pretrained student model — outperforms all other methods across the board. Even more interestingly, on Amazon-5 and Yelp-5, TGT with randomly initialized student, i.e., TGT (Random Init), outperforms the standard approach of finetuning a pretrained model with one-hot labels, i.e., One-hot (Pretrained).

distribution modeled by the generator is not very large, which is the case for ImageNet-LT. The fact that TGT (random) (cf. Eq. (4)) provides large gains over standard distillation establishes the value of utilizing the latent space, as suggested by our analysis in Section 3.3. Note that TGT (gradient-based) brings further gains over TGT (random), particularly on SUN-LT and Places-LT which are extremely long-tail. We believe that gradient-based first-order exploration is specifically useful for settings where data is extremely sparse or where isotropic random perturbation in the latent space does not produce diverse enough instances. A systematic study of this constitutes an interesting avenue for future research. Owing to its computational efficiency, we focus on TGT (random) for rest of paper.

Note that some of the baselines in Table 1 rely on specialized loss functions and/or training methods designed for long-tail settings, whereas we do not leverage such techniques. Combining the TGT framework with a long-tail specific loss function as opposed to using the standard cross-entropy loss function can potentially improve its performance. We leave this direction for future explorations.

## 4.2 TGT IN LOW-DATA REGIME

To further showcase effectiveness of knowledge transfer via TGT, we simulate a low-data regime by varying the amount of available training data for ImageNet (Russakovsky et al., 2015) and studying its impact on student's performance. We use the same model architectures as in Section 4.1, but finetune the teacher labeler on the entire ImageNet. We then compare the performance of the student trained via TGT, with the students trained via normal training (with one-hot labels) and standard distillation.

Fig. 2 shows that both TGT and standard distillation utilize additional training data more effectively than normal training, with TGT being the most efficient of the two. Interestingly, employing TGT is **equivalent to an increase in sample size by 4x**, compared to the normal training. This verifies that TGT generates informative training instances for the student.

## 4.3 TEXT CLASSIFICATION

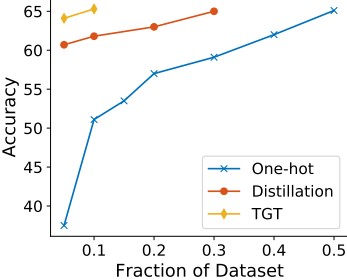

Figure 2: Comparison among normal training (one-hot), standard distillation (distillation), and TGT in simulated low-data regimes. We imitate a low-data regime via subsampling the ImageNet training set and evaluate the trained student models on the entire eval set. We employ 450k training steps for normal training and standard distillation, and 112k training steps for TGT. TGT outperforms other methods in less training steps, thus, effectively simulating an increased sample size.

**Setup.** We evaluate the proposed TGT framework on four benchmark text classification datasets: Amazon-5 (Zhang et al., 2015), IMDB (Maas et al., 2011), MNLI (Williams et al., 2018), and Yelp-5 (Zhang et al., 2015). Following Xie et al. (2020a), we also consider an extremely sub-sampled version of Amazon-5 and Yelp-5 consisting of only 2.5k labeled examples. Again, we utilize off-the-

shelf teacher models, in particular a BART-base (Lewis et al., 2020) and RoBERTa-large (Liu et al., 2019a) as the teacher generator and teacher labeler, respectively. Following Rashid et al. (2021), we employ a DistilBERT (Sanh et al., 2019b) model as student architecture. Both teacher networks are pretrained on a very large generic text corpus of size 160GB. The teacher labeler is finetuned on each task-specific dataset while the teacher generator is not specialized to any specific task.

**Results.** We compare TGT with other data augmentation and distillation baselines in Table 2. Note that TGT considerably improves the performance and beats the state-of-the-art methods MATE-KD (Rashid et al., 2021) and UDA (Xie et al., 2020a). Interestingly, by using TGT on a *randomly initialized* student, we can match the performance of finetuning (with one-hot labels) a *pretrained* model on Amazon-5 and Yelp-5. We highlight that baselines such as MATE-KD always work with a pretrained student model. Thus, the improvements realized by TGT with a randomly initialized student demonstrates enormous saving in overall data and training time requirement as it eliminates the need for pretraining on a large corpus. This further establishes that TGT can enable a *data-efficient knowledge transfer* from the teacher to the student.

## 4.4 TEXT RETRIEVAL

**Setup.** Finally, we evaluate TGT on Natural Questions (NQ) (Kwiatkowski et al., 2019) — a text retrieval benchmark. The task is to find a matching passage for a question, out of a large candidate passage corpus (21M). We use RoBERTa-Base dual-encoder model Oğuz et al. (2021) as teacher labeler and BART-base (Lewis et al., 2020) as teacher generator. We utilize DistilBERT dual encoder model as our student architecture. We follow the standard retrieval distillation setup where the teacher labeler provides labels for all the within-batch question-to-passage pairs for the student to match.

Besides one-hot training and standard distillation, we consider another baseline, namely *uniform negatives*. In uniform negatives, for each question-to-passage pair in NQ, we *uniformly* sample 2 additional passages from the passage corpus during training. TGT instead dynamically generates 2 *confusing* passages for each question-passage pair with BART generator, infusing the isotropic perturbation as per Eq. (4).

| Method | recall@20 | recall@100 |
|---|---|---|
| *Teacher (labeler) model* | 0.7957 | 0.8855 |
| One-hot | 0.6453 | 0.8198 |
| Distillation *(standard)* | 0.7486 | 0.8608 |
| Uniform negatives | 0.7536 | 0.8496 |
| TGT (**ours**) | **0.7659** | **0.8763** |

Table 3: Performance of TGT and various baselines on the NQ retrieval task. (Kwiatkowski et al., 2019). The teacher labeler follows the setup of (Oğuz et al., 2021) that pretrains RoBERTa-base on a large corpus and also PAQ (Lewis et al., 2021) and then finetuned to NQ (Kwiatkowski et al., 2019). BART-base (Lewis et al., 2020) is employed to serve as a task-agnostic generator. All student models follow the architecture of DistilBERT(Sanh et al., 2019b). TGT significantly outperforms standard training (One-hot) and teacher-label only distillation (Distillation). TGT closes the teacher-student gap by *37%* at @20, *63%* at @100) compared to the standard distillation. See Appendix F.4 for more details on the experimental setup.

**Results.** Table 3 shows that TGT significantly improves performance, closing the teacher-student gap by *37%* at recall@20 and *63%* at recall@100 compared to the standard distillation. Unlike TGT, uniform negatives only partially helped (slight improvement on recall@20 but degradation one recall@100 compared to the standard distillation). A plausible explanation is that, due to the extremely large passage corpus (21M), uniformly sampled passages are not very relevant to the matching question-to-passage pair in NQ. TGT instead generates informative passages that are close to the matching pair.

## 5 CONCLUSION AND FUTURE DIRECTIONS

We have introduced a simple and theoretically justified distillation scheme (TGT) that adaptively generates samples with the aim of closing the divergence between student and teacher predictions. Our results show it to outperform, in aggregate, existing distillation approaches. Unlike alternative methods, it is also applicable to both continuous and discrete domains, as the results on image and text data show. TGT is orthogonal to other approaches that enable efficient inference such as quantization and pruning, and combining them is an interesting avenue for future work. Another potential research direction is to employ TGT for multi-modal data which would require accommodating multiple generative models with their own latent spaces, raising both practical and theoretical challenges.

## ETHICS STATEMENT

TGT framework relies on the availability of a good-quality teacher for the underlying domain to provide efficient distillation. The impact of knowledge distillation on transferring the teacher model's biases to the resulting student model is far from well understood. Moreover, the teacher generator that TGT utilizes are often large pretrained models trained on lot of unfiltered data. As a result these large models can have potential biases without the awareness of the user. Also, how various biases present in the generator impact the student model's fairness/bias is not addressed in our work. A deeper study of this issue is required for our proposed method, as well as for the knowledge distillation as an ML technique in general.

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

## A    FURTHER COMPARISON WITH MATE-KD

MATE-KD (Rashid et al., 2021) alternative trains generator model and student model, with the hope of generating most adversarial examples for the students during the training. This can cause the generator to drift away from true data distribution. In contrast, we keep the pre-trained teacher-generator model fixed throughout the training process of the student. Our objective behind employing the generator model is to leverage the domain knowledge it has already acquired during its pre-training. While we do want to generate 'hard instances' for the student, we also want those instances to be relevant for the underlying task. Thus, keeping the generator fixed introduces a regularization where the training instances the student encounters do not introduce domain mismatch.

Keeping in mind the objective of producing new informative training instances that are in-domain, we introduce perturbation in the latent space realized by the encoder of the teacher-generator model (see Figure 1). This is different from directly perturbing an original training instance in the input space itself, as done by MATE-KD. As evident from our theoretical analysis and empirical evaluation, for a fixed teacher-generator model, employing perturbation in the latent space leads to more informative data augmentation and enables good performance on both image and text domain.

## B    BACKGROUND AND NOTATION

For $a, b \in \mathbb{R}$, we use $a = \mathcal{O}(b)$ to denote that there exits a constant $\gamma > 0$ such that $a \leq \gamma \cdot b$.

Given a collection of $n$ i.i.d. random variables $\mathcal{U}_n = \{u_1, \ldots, u_n\} \subset \mathcal{U}$, generated from a distribution $\mathcal{D}_U$ and a function $\tau : \mathcal{U} \to \mathbb{R}$, we define the *empirical mean* of $\{\tau(u_1), \ldots, \tau(u_n)\}$ as

$$\mathbb{E}_{\mathcal{U}_n}[\tau(U)] := \frac{1}{n} \sum_{i \in [n]} \tau(u_i). \tag{7}$$

For the underlying multiclass classification problem defined by the distribution $\mathcal{D} := \mathcal{D}_{X \times Y}$, we assume that the label set $\mathcal{Y}$ with $K$ classes takes the form $[K] := \{1, \ldots, K\}$. We use $\mathcal{F}$ to denote the collection of potential classification models that the learning methods is allowed to select from, namely *function class* or *hypothesis set*:

$$\mathcal{F} \subseteq \{\mathcal{X} \to \mathbb{R}^K\}, \tag{8}$$

which is a subset of all functions that map elements of the instance space $\mathcal{X}$ to the elements of $\mathbb{R}^K$.

Given a classification loss function $\ell : \mathbb{R}^K \times \mathcal{Y} \to \mathbb{R}$ and a model $f : \mathcal{X} \to \mathbb{R}^K$ and a sample $\mathcal{S}_n^{\text{labeled}} = \{(x_i, y_i)\}_{i \in [n]}$ generated from $\mathcal{D}$, we define the *empirical risk* for $f \in \mathcal{F}$ as follows.

$$R_{\ell,f}(\mathcal{S}_n^{\text{labeled}}) := \mathbb{E}_{\mathcal{S}_n^{\text{labeled}}}[\ell(f(X))] = \frac{1}{n} \sum_{i \in [n]} \ell(f(x_i), y_i). \tag{9}$$

Further, we define the *population* risk for $f \in \mathcal{F}$ associated with data distribution $\mathcal{D}$ as follows.

$$R_{\ell,f}(\mathcal{D}) = \mathbb{E}_{X,Y \sim \mathcal{D}}[\ell(f(X), Y)]. \tag{10}$$

Note that, when the loss function $\ell$ is clear from the context, we drop $\ell$ from the notation and simply use $R_f(\mathcal{S}_n^{\text{labeled}})$ and $R_f(\mathcal{D})$ to denote the the empirical and populations risks for $f$, respectively.

Given a function class $\mathcal{F}$, the loss function $\ell$ induces the following function class.

$$\mathcal{G}_{\ell,\mathcal{F}} = \{(x, y) \mapsto \ell(f(x), y) : f \in \mathcal{F}\}. \tag{11}$$

**Definition B.1** (Rademacher complexity of $\mathcal{G}_{\ell,\mathcal{F}}$). Now, given a sample $\mathcal{S}_n^{\text{labeled}} = \{(x_i, y_i)\}_{i \in [n]} \sim \mathcal{D}^n$ and a vector $\boldsymbol{\sigma} = (\sigma_i, \ldots, \sigma_m) \in \{+1, -1\}$ with $n$ i.i.d. Bernoulli random variables, empirical Rademacher complexity $\mathfrak{R}_{\mathcal{S}}(\mathcal{G}_{\ell,\mathcal{F}})$ and Rademacher complexity $\mathfrak{R}_n(\mathcal{G}_{\ell,\mathcal{F}})$ are defined as

$$\mathfrak{R}_{\mathcal{S}_n^{\text{labeled}}}(\mathcal{G}_{\ell,\mathcal{F}}) = \frac{1}{n} \mathbb{E}_{\boldsymbol{\sigma}} \left[ \sup_{g \in \mathcal{G}_{\ell,\mathcal{F}}} \sum_{i=1}^n \sigma_i g(x_i, y_i) \right]$$

and

$$\mathfrak{R}_n(\mathcal{G}_{\ell,\mathcal{F}}) = \mathbb{E}_{\mathcal{S} \sim \mathcal{D}^n} \left[ \mathfrak{R}_{\mathcal{S}_n^{\text{labeled}}}(\mathcal{G}_{\ell,\mathcal{F}}) \right] \tag{12}$$

Let $S_n = \{x_i\}_{i \in [n]}$ be a set of $n$ *unlabeled* samples generated from $\mathcal{D}_X$. Then, given a teacher model $h : \mathcal{X} \to \mathbb{R}^K$ and a distillation loss $\ell_{\mathrm{d}} : \mathbb{R}^K \times \mathbb{R}^K \to \mathbb{R}$, we define the *empirical (distillation) risk* for $f \in \mathcal{F}$ to be

$$R_{\ell_{\mathrm{d}}, f}^h(S_n) := \mathbb{E}_{S_n}\left[\ell_{\mathrm{d}}(f(X), h(X))\right] = \frac{1}{n}\sum_{i \in [n]} \ell_{\mathrm{d}}\big(f(x_i), h(x_i)\big). \tag{13}$$

Accordingly, the *population (distillation) risk* for $f \in \mathcal{F}$ is defined as

$$R_{\ell_{\mathrm{d}}, f}^h(\mathcal{D}) := \mathbb{E}_{X \sim \mathcal{D}_X}\left[\ell_{\mathrm{d}}(f(X), h(X))\right]. \tag{14}$$

Again, when $\ell_{\mathrm{d}}$ is clear from the context, we simply use $R_f^h(S_n)$ and $R_f^h(\mathcal{D})$ to denote the empirical and population distillation risk for $f$, respectively.

Note that, for a (student) function class $\mathcal{F}$ and a teacher model $h$, $\ell_{\mathrm{d}}$ produces an induced function class $\mathcal{G}_{\ell_{\mathrm{d}}, h}(\mathcal{F})$, defined as

$$\mathcal{G}_{\ell_{\mathrm{d}}, \mathcal{F}}^h := \{x \mapsto \ell_{\mathrm{d}}(f(x), h(x)) : f \in \mathcal{F}\}. \tag{15}$$

**Definition B.2** (Rademacher complexity of $\mathcal{G}_{\ell_{\mathrm{d}}, \mathcal{F}}^h$). Given a sample $S_n = \{x_i\}_{i \in [n]} \sim \mathcal{D}_X^n$ and a vector $\boldsymbol{\sigma} = (\sigma_i, \ldots, \sigma_m) \in \{+1, -1\}$ with $n$ i.i.d. Bernoulli randoms variable, empirical Rademacher complexity $\mathfrak{R}_{S_n}\big(\mathcal{G}_{\ell_{\mathrm{d}}, \mathcal{F}}^h\big)$ and Rademacher complexity $\mathfrak{R}_n\big(\mathcal{G}_{\ell_{\mathrm{d}}, \mathcal{F}}^h\big)$ are defined as

$$\mathfrak{R}_{S_n}(\mathcal{G}_{\ell_{\mathrm{d}}, \mathcal{F}}^h) = \frac{1}{n}\mathbb{E}_{\boldsymbol{\sigma}}\left[\sup_{g \in \mathcal{G}_{\ell_{\mathrm{d}}, \mathcal{F}}^h} \sum_{i=1}^n \sigma_i g(x_i)\right], \tag{16}$$

and

$$\mathfrak{R}_n(\mathcal{G}_{\ell_{\mathrm{d}}, \mathcal{F}}^h) = \mathbb{E}_{S \sim \mathcal{D}_X^n}\left[\mathfrak{R}_{S_n}(\mathcal{G}_{\ell_{\mathrm{d}}, \mathcal{F}}^h)\right] \tag{17}$$

respectively.

## C    DEFERRED PROOFS FROM SECTION 3

### C.1    PROOF OF THEOREM 3.1

In this subsection, we present a general version of Theorem 3.1. Before that, we state the following relevant assumption on the distillation loss $\ell_{\mathrm{d}}$.

**Assumption C.1.** Let $\ell : \mathbb{R}^K \times \mathcal{Y} \to \mathbb{R}$ be a bounded loss function. For a teacher function $h : \mathcal{X} \to \mathbb{R}^K$, the distillation loss $\ell_{\mathrm{d}}$ takes the form

$$\ell_{\mathrm{d}}(f(x), h(x)) = \sum_{y \in [K]} h(x)_y \cdot \ell(f(x), y).$$

*Remark* C.2. Note that the cross-entropy loss $\ell_{\mathrm{d}}(f(x), h(x)) = -\sum_y h(x)_y \cdot \log\big(f(x)_y\big)$, here, one of the most common choices for the distillation loss, indeed satisfies Assumption C.1.[2]

The following results is a general version of Theorem 3.1 in the main body.

**Theorem C.3.** *Let a generator with the encoder* Enc *and decoder* Dec *ensures the approximation guarantee in Eq.* (6) *for* $\mathcal{D}_X$. *Let* Dec *and teacher labeler be Lipschtiz functions,* $\mathcal{F}$ *be function class of Lipschitz functions, and the distillation loss* $\ell_{\mathrm{d}}$ *be Lipschtiz. Then, with probability at least* $1 - \delta$, *the following holds for any* $f \in \mathcal{F}$.

$$R_{\ell_{\mathrm{d}}, f}^h(\mathcal{D}_X) \le R_{\ell_{\mathrm{d}}, f}^h(S_n) + \mathcal{O}\big(n^{-1/d}\big) + L\epsilon + \mathcal{O}\Big(\sqrt{\frac{\log(1/\delta)}{n}}\Big), \tag{18}$$

---

[2]For the sake of brevity, we simply include the softmax-operation in the definition of $h$ and $f$, i.e., $h(x)$ and $f(x)$ are valid probability distributions over $\mathcal{Y} = [K]$.

where $L$ denotes the effective Lipschitz constant of the induced function class $\mathcal{G}^h_{\ell_d,\mathcal{F}}$ in Eq. (15). Additionally, if the distillation loss $\ell_d$ satisfies Assumption C.1 with a classification loss $\ell$, then Eq. (18) further implies the following.

$$R_{\ell,f}(\mathcal{D}) \le R^h_{\ell_d,f}(\mathcal{S}_n) + \mathcal{O}(n^{-1/d}) + L\epsilon + \mathcal{O}\left(\sqrt{\frac{\log(1/\delta)}{n}}\right) + \mathcal{O}\left(\sqrt{K} \cdot \mathbb{E}_{\mathcal{D}_X}\left[\|\mathcal{D}_{Y|X} - h(X)\|_2\right]\right).$$
(19)

*Proof.* Note that

$$R^h_{\ell_d,f}(\mathcal{D}_X) = \mathbb{E}_{\mathcal{D}_X}[\ell_d(f(X), h(X))]$$

$$\le \mathbb{E}_{\mathcal{S}_n}[\ell_d(f(X), h(X))] + \sup_{f \in \mathcal{F}} \left| \mathbb{E}_{\mathcal{S}_n}[\ell_d(f(X), h(X))] - \mathbb{E}_{\mathcal{D}_X}[\ell_d(f(X), h(X))] \right|$$

$$\overset{(i)}{\le} \mathbb{E}_{\mathcal{S}_n}[\ell_d(f(X), h(X))] + \sup_{g \in \mathcal{G}^h_{\ell_d,\mathcal{F}}} \left| \mathbb{E}_{\mathcal{S}_n}[g(X)] - \mathbb{E}_{\mathcal{D}_X}[g(X)] \right|$$

$$\overset{(ii)}{\le} \mathbb{E}_{\mathcal{S}_n}[\ell_d(f(X), h(X))] + \mathfrak{R}_{\mathcal{S}_n}(\mathcal{G}^h_{\ell_d,\mathcal{F}}),$$
(20)

where $(i)$ follows from the definition of $\mathcal{G}^h_{\ell_d,\mathcal{F}}$ in Eq. (15) and $(i)$ follow from the standard symmetrization argument (Devroye et al., 2013; Mohri et al., 2018). Next, we turn our focus to the empirical Rademacher complexity $\mathfrak{R}_{\mathcal{S}_n}(\mathcal{G}^h_{\ell_d,\mathcal{F}})$. Recall that $\mathcal{S}_n = \{x_1, x_2, \dots, x_n\}$ contains $n$ i.i.d. samples generated from the distribution $\mathcal{D}_X$. We define another set of $n$ points

$$\tilde{\mathcal{S}}_n = \{\tilde{x}_1 = \mathsf{Dec} \circ \mathsf{Enc}(x_1), \dots, \tilde{x}_n = \mathsf{Dec} \circ \mathsf{Enc}(x_n)\}.$$

It follows from our assumption on the quality of the generator (cf. Eq. (6)) that

$$\|\mathsf{Dec} \circ \mathsf{Enc}(x_i) - x_i\| \le \epsilon, \quad \forall i \in [n].$$
(21)

Note that

$$\mathfrak{R}_{\mathcal{S}_n}(\mathcal{G}^h_{\ell_d,\mathcal{F}}) = \frac{1}{n} \mathbb{E}_{\boldsymbol{\sigma}} \left| \sup_{g \in \mathcal{G}^h_{\ell_d,\mathcal{F}}} \sum_i \sigma_i g(x_i) \right|,$$

where $\boldsymbol{\sigma}$ denote a vector with $n$ i.i.d Bernoulli random variables.

$$\mathfrak{R}_{\mathcal{S}_n}(\mathcal{G}^h_{\ell_d,\mathcal{F}}) = \frac{1}{n} \mathbb{E}_{\boldsymbol{\sigma}} \left| \sup_{g \in \mathcal{G}^h_{\ell_d,\mathcal{F}}} \frac{1}{n} \sum_i \sigma_i \big(g(\tilde{x}_i) - g(\tilde{x}_i) + g(x_i)\big) \right|$$

$$\le \frac{1}{n} \mathbb{E}_{\boldsymbol{\sigma}} \left| \sup_{g \in \mathcal{G}^h_{\ell_d,\mathcal{F}}} \frac{1}{n} \sum_i \sigma_i g(\tilde{x}_i) \right| +$$

$$\frac{1}{n} \mathbb{E}_{\boldsymbol{\sigma}} \left| \sup_{g \in \mathcal{G}^h_{\ell_d,\mathcal{F}}} \sum_i \sigma_i \big(g(x_i) - g(\tilde{x}_i)\big) \right|$$

$$\le \frac{1}{n} \mathbb{E}_{\boldsymbol{\sigma}} \left| \sup_{g \in \mathcal{G}^h_{\ell_d,\mathcal{F}}} \sum_i \sigma_i g(\tilde{x}_i) \right| + \sup_{g \in \mathcal{G}^h_{\ell_d,\mathcal{F}}} \frac{1}{n} \sum_i |g(x_i) - g(\tilde{x}_i)|$$

$$\le \frac{1}{n} \mathbb{E}_{\boldsymbol{\sigma}} \left| \sup_{g \in \mathcal{G}^h_{\ell_d,\mathcal{F}}} \sum_i \sigma_i g(\tilde{x}_i) \right| + \frac{1}{n} \sum_i L \cdot \|x_i - \tilde{x}_i\|$$

$$\le \frac{1}{n} \mathbb{E}_{\boldsymbol{\sigma}} \left| \sup_{g \in \mathcal{G}^h_{\ell_d,\mathcal{F}}} \sum_i \sigma_i g(\tilde{x}_i) \right| + L\epsilon$$

$$\le \frac{1}{n} \mathbb{E}_{\boldsymbol{\sigma}} \left| \sup_{g \in \mathcal{G}^h_{\ell_d,\mathcal{F}}} \sum_i \sigma_i g(\mathsf{Dec}(z_i)) \right| + L\epsilon,$$
(22)

where $z_i = \mathsf{Enc}(x_i)$, for $i \in [n]$. By definition of $\mathcal{G}^h_{\ell_\mathrm{d},\mathcal{F}}$, $g(\mathsf{Dec}(e)) = \ell_\mathrm{d}(f(x), h(x))$ for some $f \in \mathcal{F}$. Now, we can define a new function class from $\mathbb{R}^d$ to $\mathbb{R}$:

$$\mathcal{G}^{h,\mathsf{Dec}}_{\ell_\mathrm{d},\mathcal{F}} = \{z \mapsto \ell_\mathrm{d}(f \circ \mathsf{Dec}(z), h \circ \mathsf{Dec}(z)) : f \in \mathcal{F}\}. \tag{23}$$

Therefore, it follows from Eq. (22) and Eq. (23) that

$$\mathfrak{R}_{\mathcal{S}_n}(\mathcal{G}^h_{\ell_\mathrm{d},\mathcal{F}}) \le \mathfrak{R}_{\mathcal{E}_n}(\mathcal{G}^{h,\mathsf{Dec}}_{\ell_\mathrm{d},\mathcal{F}}) + L\epsilon, \tag{24}$$

where $\mathcal{E}_n = \{\mathsf{Enc}(x_1), \ldots, \mathsf{Enc}(x_n)\} \subset \mathbb{R}^d$. It follows from the standard concentration results for *empirical* Rademacher complexity around Rademacher complexity that with probability at least $1 - \delta$,

$$\mathfrak{R}_{\mathcal{E}_n}(\mathcal{G}^{h,\mathsf{Dec}}_{\ell_\mathrm{d},\mathcal{F}}) \le \mathfrak{R}_n(\mathcal{G}^{h,\mathsf{Dec}}_{\ell_\mathrm{d},\mathcal{F}}) + \mathcal{O}\Big(\sqrt{\log\Big(\frac{1}{\delta}\Big) \cdot \frac{1}{n}}\Big). \tag{25}$$

Since $f \in \mathcal{F}$, $h$ and $\mathsf{Dec}$ are Lipschitz functions, $\mathcal{G}^{h,\mathsf{Dec}}_{\ell_\mathrm{d},\mathcal{F}}$ is collection of Lipschitz functions from $\mathbb{R}^d$ to $\mathbb{R}$. Thus, it follows from the standard results (Gottlieb et al., 2016, Theorem 4.3) that

$$\mathfrak{R}_n(\mathcal{G}^{h,\mathsf{Dec}}_{\ell_\mathrm{d},\mathcal{F}}) \le \mathcal{O}\big(n^{-\frac{1}{d}}\big). \tag{26}$$

Now, Eq. (18) follow from Eq. (20), Eq. (24), Eq. (25), and Eq. (26). Finally, Eq. (19) follows by combining Lemma D.4 with Eq. (18). $\qquad\square$

## C.2 PROOF OF THEOREM 3.2

Here, we present the following result, which along with Theorem C.5 implies Theorem 3.2 stated in the main body.

**Theorem C.4.** *Let $\tilde{\mathcal{S}}_n = \{\tilde{x}_i\}_{i \in [n]}$ be $n$ i.i.d. samples generated from a distribution $\tilde{\mathcal{D}}_X$. Further, let $\tilde{f}_n \in \mathcal{F}$ denote the student model learned via distillation on $\tilde{\mathcal{S}}_n$, with $h$ and $\ell_\mathrm{d}$ as the teacher model and distillation loss, respectively. Then, with probability at least $1 - \delta$, we have*

$$R^h_{\ell_\mathrm{d},\tilde{f}_n}(\mathcal{D}_X) \le R^h_{\ell_\mathrm{d},\tilde{f}_n}(\tilde{\mathcal{S}}_n) + \mathcal{W}(\mathcal{D}_X, \tilde{\mathcal{D}}_X) + \tilde{\mathfrak{R}}_n(\mathcal{G}^h_{\ell_\mathrm{d},\mathcal{F}}) + \mathcal{O}\Big(\sqrt{\log\Big(\frac{1}{\delta}\Big) \cdot \frac{1}{n}}\Big), \tag{27}$$

*where $\tilde{\mathfrak{R}}_n(\mathcal{G}^h_{\ell_\mathrm{d},\mathcal{F}}) = \mathbb{E}_{\tilde{\mathcal{S}} \sim \tilde{\mathcal{D}}^n}\Big[\mathfrak{R}_{\tilde{\mathcal{S}}_n}(\mathcal{G}^h_{\ell_\mathrm{d},\mathcal{F}})\Big]$ denote that Rademacher complexity of the induced function class $\mathcal{G}^h_{\ell_\mathrm{d},\mathcal{F}}$, defined in Eq. (15). If $\tilde{\mathcal{S}}$ is constructed with the TGT framework based on a generator with the encoder $\mathsf{Enc}$ and decoder $\mathsf{Dec}$, then Eq. (27) further specialized to*

$$R^h_{\ell_\mathrm{d},\tilde{f}_n}(\mathcal{D}_X) \le R^h_{\ell_\mathrm{d},\tilde{f}_n}(\tilde{\mathcal{S}}_n) + \mathcal{W}(\mathcal{D}_X, \tilde{\mathcal{D}}_X) + \tilde{\mathfrak{R}}_n(\mathcal{G}^{h,\mathsf{Dec}}_{\ell_\mathrm{d},\mathcal{F}}) + \mathcal{O}\Big(\sqrt{\log\Big(\frac{1}{\delta}\Big) \cdot \frac{1}{n}}\Big), \tag{28}$$

*where $\mathcal{G}^{h,\mathsf{Dec}}_{\ell_\mathrm{d},\mathcal{F}}$ defines the following induced function class from $\mathbb{R}^d$ (i.e., the latent space of the generator) to $\mathbb{R}$.*

$$\mathcal{G}^{h,\mathsf{Dec}}_{\ell_\mathrm{d},\mathcal{F}} = \{z \mapsto \ell_\mathrm{d}(f \circ \mathsf{Dec}(z), h \circ \mathsf{Dec}(z)) : f \in \mathcal{F}\}. \tag{29}$$

*Proof.* Note that

$$
\begin{aligned}
R^h_{\ell_\mathrm{d},\tilde{f}_n}(\tilde{\mathcal{D}}_X) &= \mathbb{E}_{\tilde{\mathcal{D}}_X}[\ell_\mathrm{d}(\tilde{f}_n(X), h(X))] \\
&\le \mathbb{E}_{\tilde{\mathcal{S}}_n}[\ell_\mathrm{d}(\tilde{f}_n(X), h(X))] + \sup_{f \in \mathcal{F}}\Big|\mathbb{E}_{\tilde{\mathcal{S}}_n}[\ell_\mathrm{d}(f(X), h(X))] - \mathbb{E}_{\tilde{\mathcal{D}}_X}[\ell_\mathrm{d}(f(X), h(X))]\Big| \\
&\le \mathbb{E}_{\tilde{\mathcal{S}}_n}[\ell_\mathrm{d}(\tilde{f}_n(X), h(X))] + \sup_{g \in \mathcal{G}^h_{\ell_\mathrm{d},\mathcal{F}}}\Big|\mathbb{E}_{\tilde{\mathcal{S}}_n}[g(X)] - \mathbb{E}_{\tilde{\mathcal{D}}_X}[g(X)]\Big| \\
&\le \mathbb{E}_{\tilde{\mathcal{S}}_n}[\ell_\mathrm{d}(\tilde{f}_n(X), h(X))] + \mathfrak{R}_{\tilde{\mathcal{S}}_n}(\mathcal{G}^h_{\ell_\mathrm{d},\mathcal{F}}),
\end{aligned} \tag{30}
$$

where the last two inequality follows from the definition of $\mathcal{G}^h_{\ell_\mathrm{d},\mathcal{F}}$ (cf. Eq. (15)) and the standard symmetrization argument (Devroye et al., 2013; Mohri et al., 2018), respectively.

Now, the standard concentration results for empirical Rademacher complexity implies that, with probability at least $1 - \delta$, we have the following.

$$\mathfrak{R}_{\tilde{\mathcal{S}}_n}(\mathcal{G}^h_{\ell_d, \mathcal{F}}) \leq \mathbb{E}_{\tilde{\mathcal{S}} \sim \tilde{\mathcal{D}}^n}\left[\mathfrak{R}_{\tilde{\mathcal{S}}_n}(\mathcal{G}^h_{\ell_d, \mathcal{F}})\right] + \mathcal{O}\left(\sqrt{\log\left(\frac{1}{\delta}\right) \cdot \frac{1}{n}}\right) \tag{31}$$

$$= \tilde{\mathfrak{R}}_n(\mathcal{G}^h_{\ell_d, \mathcal{F}}) + \mathcal{O}\left(\sqrt{\log\left(\frac{1}{\delta}\right) \cdot \frac{1}{n}}\right). \tag{32}$$

It follows from Lemma D.3 that

$$R^h_{\ell_d, \tilde{f}_n}(\mathcal{D}_X) \leq R^h_{\ell_d, \tilde{f}_n}(\tilde{\mathcal{D}}_X) + \mathcal{W}(\mathcal{D}_X, \tilde{\mathcal{D}}_X) \tag{33}$$

Now the first part of Theorem C.4, as stated in Eq. (27), follows by combining Eq. (30), Eq. (31), and Eq. (33).

We now focus on establishing Eq. (28). Note that, for a sample $\tilde{\mathcal{S}}_n = \{\tilde{x}_1, \ldots, \tilde{x}_n\}$ generated by the TGT framework, there exists $\{z_1, \ldots, z_n\} \subset \mathbb{R}^d$ such that

$$\tilde{x}_i = \mathsf{Dec}(z_i), \ \forall i \in [n]. \tag{34}$$

Thus,

$$\begin{aligned}
\mathfrak{R}_{\tilde{\mathcal{S}}_n}(\mathcal{G}^h_{\ell_d, \mathcal{F}}) &= \frac{1}{n}\mathbb{E}_{\boldsymbol{\sigma}}\left|\sup_{g \in \mathcal{G}^h_{\ell_d, \mathcal{F}}} \sum_i \sigma_i g(\tilde{x}_i)\right| \\
&\overset{(i)}{=} \frac{1}{n}\mathbb{E}_{\boldsymbol{\sigma}}\left|\sup_{g \in \mathcal{G}^h_{\ell_d, \mathcal{F}}} \sum_i \sigma_i g(\mathsf{Dec}(z_i))\right| \\
&\leq \frac{1}{n}\mathbb{E}_{\boldsymbol{\sigma}}\left|\sup_{g' \in \mathcal{G}^{h, \mathsf{Dec}}_{\ell_d, \mathcal{F}}} \sum_i \sigma_i g'(z_i)\right| \\
&= \mathfrak{R}_{\tilde{\mathcal{S}}_n}(\mathcal{G}^{h, \mathsf{Dec}}_{\ell_d, \mathcal{F}}),
\end{aligned} \tag{35}$$

where $(i)$ employs Eq. (34). Thus, combining Eq. (30) and Eq. (35) gives us that

$$R^h_{\ell_d, \tilde{f}_n}(\tilde{\mathcal{D}}_X) \leq \mathbb{E}_{\tilde{\mathcal{S}}_n}[\ell_d(\tilde{f}_n(X), h(X))] + \mathfrak{R}_{\tilde{\mathcal{S}}_n}(\mathcal{G}^{h, \mathsf{Dec}}_{\ell_d, \mathcal{F}}). \tag{36}$$

Now, similar to the proof of Eq. (27), we can invoke Lemma D.3 and the concentration result for empirical Rademacher complexity to obtain the desired result in Eq. (28) from Eq. (36). $\quad\square$

*Remark* C.5. Note that, if the distillation loss $\ell_d$ satisfies Assumption C.1 with a loss function $\ell$, then, one can combine Theorem C.4 and Lemma D.4 to readily obtain bounds on $R_{\ell, \tilde{f}_n}(\mathcal{D})$ with an additional term

$$\mathcal{O}\left(\sqrt{K} \cdot \mathbb{E}_{\mathcal{D}_X}\left[\|\mathcal{D}_{Y|X} - h(X)\|_2\right]\right).$$

This term captures the quality of the teacher labeler $h$.

## C.3 Weighted ERM: An alternative training procedure for TGT

Note that given the samples $\tilde{\mathcal{S}}_n = \{\tilde{x}_i\}_{i \in [n]}$ generated from $\tilde{\mathcal{D}}_X$ and a teacher labeler $h$, we minimize the following empirical risk for student training:

$$R^h_{\ell_d, f}(\tilde{\mathcal{S}}_n) = \frac{1}{n}\sum_{i \in [n]} \ell_d\big(f(\tilde{x}_i), h(\tilde{x}_i)\big). \tag{37}$$

However, as we notice in Theorem C.4, this leads to an additional $\mathcal{W}(\mathcal{D}_X, \tilde{\mathcal{D}}_X)$ penalty term in the generalization bound. One standard approach to address this issue is to consider the following *weighted* empirical risk.

$$R^{h, \mathrm{IS}}_{\ell_d, f}(\tilde{\mathcal{S}}_n) = \frac{1}{n}\sum_{i \in [n]} \ell_d\big(f(\tilde{x}_i), h(\tilde{x}_i)\big) \cdot \frac{p_{\mathcal{D}_X}(\tilde{x}_i)}{p_{\tilde{\mathcal{D}}_X}(\tilde{x}_i)}, \tag{38}$$

where $p_{\mathcal{D}_X}$ and $p_{\tilde{\mathcal{D}}_X}$ denote the probability density function (pdf) for $\mathcal{D}_X$ and $\tilde{\mathcal{D}}_X$.[3] Accordingly, we define a new induced function class related to the weighted empirical risk:

$$\mathcal{G}_{\ell_{\mathrm{d}},\mathcal{F}}^{h\mathrm{IS}} = \left\{ x \mapsto \ell_{\mathrm{d}}\big(f(\tilde{x}_i), h(\tilde{x})\big) \cdot \frac{p_{\mathcal{D}_X}(\tilde{x})}{p_{\tilde{\mathcal{D}}_X}(\tilde{x})} : f \in \mathcal{F} \right\} \tag{39}$$

Importantly, we have

$$R_{\ell_{\mathrm{d}},f}^{h,\mathrm{IS}}(\tilde{\mathcal{D}}_X) = \mathbb{E}_{\tilde{\mathcal{D}}_X}\left[ R_{\ell_{\mathrm{d}},f}^{h,\mathrm{IS}}(\tilde{\mathcal{S}}_n) \right] = R_{\ell_{\mathrm{d}},f}^{h}(\mathcal{D}_X) \tag{40}$$

Thus, following the analysis utilized in Theorem C.4, one can obtain a high probability generalization of the form.

$$R_{\ell_{\mathrm{d}},f}^{h}(\mathcal{D}_X) \leq R_{\ell_{\mathrm{d}},f}^{h,\mathrm{IS}}(\tilde{\mathcal{S}}_n) + \tilde{\mathfrak{R}}_n(\mathcal{G}_{\ell_{\mathrm{d}},\mathcal{F}}^{h,\mathrm{IS}}) + \mathcal{O}\left( \sqrt{\log\left(\frac{1}{\delta}\right) \cdot \frac{1}{n}} \right), \tag{41}$$

which avoids the $\mathcal{W}(\mathcal{D}_X, \tilde{\mathcal{D}}_X)$ term.

In what follows, we explore an alternative approach to highlight the importance of the sampling approach adapted by (gradient-based) TGT. By leveraging the variance-based generalization bound (Maurer & Pontil, 2009) that were previously utilized by Menon et al. (2021a) in the context distillation, we obtain the following result for the weighted empirical risk in Eq. (38).

**Proposition C.6.** *Let $h$, $\ell_{\mathrm{d}}$, $\mathcal{F}$ and $\tilde{\mathcal{S}}_n$ be as defined in the statement of Theorem C.4. Further, assume that $\ell_{\mathrm{d},f}^{h,\mathrm{IS}}(\tilde{x}) := \ell_{\mathrm{d}}\big(f(\tilde{x}_i), h(\tilde{x})\big) \cdot \frac{p_{\mathcal{D}_X}(\tilde{x})}{p_{\tilde{\mathcal{D}}_X}(\tilde{x})}$ is bounded for all $\tilde{x} \in \mathrm{supp}(\tilde{D}_X)$. Then, for any $f \in \mathcal{F}$, the following holds with probability at least $1 - \delta$.*

$$R_{\ell_{\mathrm{d}},f}^{h}(\mathcal{D}_X) \leq R_{\ell_{\mathrm{d}},f}^{h,\mathrm{IS}}(\tilde{\mathcal{S}}_n) + (\mathrm{I}), \tag{42}$$

*where* (I) *denotes*

$$\mathcal{O}\left( \sqrt{\frac{\mathrm{Var}_{\tilde{\mathcal{D}}_X}(\ell_{\mathrm{d},f}^{h,\mathrm{IS}}(\tilde{x})) \cdot \log(\frac{\mathcal{M}(n)}{\delta})}{n}} + \frac{\log(\frac{\mathcal{M}(n)}{\delta})}{n} \right).$$

*Here, $\mathcal{M}(n) = \sup_{\mathcal{S}_n \subset \mathcal{X}^n} \mathcal{N}(1/n, \mathcal{G}_{\ell_{\mathrm{d}},\mathcal{F}}^{h,\mathrm{IS}}(\mathcal{S}_n), \|\cdot\|_\infty)$, with $\mathcal{N}(\epsilon, \mathcal{G}_{\ell_{\mathrm{d}},\mathcal{F}}^{h,\mathrm{IS}}(\mathcal{S}_n), \|\cdot\|_\infty)$ denoting the covering number (Devroye et al., 2013) of the set*

$$\mathcal{G}_{\ell_{\mathrm{d}},\mathcal{F}}^{h,\mathrm{IS}}(\mathcal{S}_n) := \{(g(x_1), \ldots, g(x_n)) : g \in \mathcal{G}_{\ell_{\mathrm{d}},\mathcal{F}}^{h,\mathrm{IS}}\}.$$

*Proof.* By utilizing the uniform convergence version of Bennet's inequality and uniform bound for $\sqrt{\mathrm{Var}_{\tilde{\mathcal{S}}_n}(\ell_{\mathrm{d}}^{\mathrm{IS}}(\tilde{x}))}$, where $\mathrm{Var}_{\tilde{\mathcal{S}}_n}(\ell_{\mathrm{d}}^{\mathrm{IS}}(\tilde{x}))$ denotes the empirical variance of $\ell_{\mathrm{d}}^{\mathrm{IS}}(\tilde{x})$ based on $\tilde{\mathcal{S}}_n$, the following holds with probability at least $1 - \delta$ (Maurer & Pontil, 2009).

$$R_{\ell_{\mathrm{d}},f}^{h,IS}(\tilde{\mathcal{D}}_X) \leq R_{\ell_{\mathrm{d}},f}^{h,\mathrm{IS}}(\tilde{\mathcal{S}}_n) + \mathcal{O}\left( \sqrt{\frac{\mathrm{Var}_{\tilde{\mathcal{D}}_X}(\ell_{\mathrm{d}}^{\mathrm{IS}}(\tilde{x})) \cdot \log(\frac{\mathcal{M}(n)}{\delta})}{n}} + \frac{\log(\frac{\mathcal{M}(n)}{\delta})}{n} \right), \forall f \in \mathcal{F}. \tag{43}$$

Since, $R_{\ell_{\mathrm{d}},f}^{h,\mathrm{IS}}(\tilde{\mathcal{D}}_X) = \mathbb{E}_{\tilde{\mathcal{D}}_X}\left[ R_{\ell_{\mathrm{d}},f}^{h,\mathrm{IS}}(\tilde{\mathcal{S}}_n) \right] = R_{\ell_{\mathrm{d}},f}^{h}(\mathcal{D}_X)$, the statement of Theorem C.6 follows from Eq. (43). □

Note that by combining Eq. (42) with Theorem D.4 translate the bound on $R_{\ell_{\mathrm{d}},f}^{h}(\mathcal{D}_X)$ to a bound on $R_{\ell,f}(\mathcal{D})$ with an additional penalty term that depends on the quality of the teacher labeler $h$.

*Remark* C.7. Eq. (42) suggests general approach to select the distribution $\tilde{\mathcal{D}}_X$ that generated the training samples $\tilde{\mathcal{S}}_n$. In order to ensure small generalization gap, it is desirable that the variance term $\mathrm{Var}_{\tilde{\mathcal{D}}_X}(\ell_{\mathrm{d}}^{\mathrm{IS}}(\tilde{x}))$ is as small as possible. Note that, the distribution that minimizes this variance takes the form

$$\log p_{\tilde{\mathcal{D}}_X}(x) \propto \log \ell_{\mathrm{d}}\big(f(x), h(x)\big) + \log p_{\mathcal{D}_X}(x), \ \forall \, x \in \mathcal{X}. \tag{44}$$

---

[3]Note that the formulation assumes that $\mathcal{D}_X \ll \tilde{\mathcal{D}}_X$, i.e., $\mathcal{D}_X$ is absolutely continuous w.r.t. $\tilde{\mathcal{D}}_X$. Also, one can replace the pdf's with probability mass functions if $\mathcal{D}_X$ and $\tilde{\mathcal{D}}_X$ are discrete distributions.

This looks like the lagrangian form of Eq. (3). Interestingly, TGT framework with *gradient-based sampling* (cf. equation 5) focuses on generating samples that maximizes the right hand side $RHS$ of Eq. (44) by first taking a sample generated according to $\mathcal{D}_X$ and then perturbing it in the *latent space* to maximize the loss $\ell_{\mathrm{d}}\big(f(x), h(x)\big)$. Thus, the resulting distribution $\tilde{\mathcal{D}}_X$ has pdf that aims to approximate the variance minimizing pdf in Eq. (44).

Here it is worth pointing out that, since exact form of $p_{\tilde{\mathcal{D}}_X}(\cdot)$ and $p_{\mathcal{D}_X}(\cdot)$ is generally not available during the training, it's not straightforward to optimize the weighted risk introduced in Eq. (38). As introduced in Section 3, TGT framework optimizes the empirical risk in Eq. (37) as opposed to minimizing Eq. (38). In this case, one can obatain a variance based bound analogous to Eq. (42) that takes the form:

$$R_{\ell_{\mathrm{d}}, f}^h(\mathcal{D}_X) \leq R_{\ell_{\mathrm{d}}, f}^h(\tilde{\mathbb{S}}_n) + \text{(II)} + \mathcal{W}(\mathcal{D}_X, \tilde{\mathcal{D}}_X), \tag{45}$$

where, (II) denotes

$$\mathcal{O}\bigg( \sqrt{\frac{\mathrm{Var}_{\tilde{\mathcal{D}}_X}(\ell_{\mathrm{d}, f}^h(\tilde{x})) \cdot \log(\frac{\mathcal{M}(n)}{\delta})}{n}} + \frac{\log(\frac{\mathcal{M}(n)}{\delta})}{n} \bigg),$$

with $\ell_{\mathrm{d}, f}^h(\tilde{x}) := \ell_{\mathrm{d}}\big(f(\tilde{x}_i), h(\tilde{x})\big)$ and $\mathcal{M}(n)$ depending the covering number for the induced function class $\mathcal{G}_{\ell_{\mathrm{d}}, \mathcal{F}}^h$ (cf. Eq. (15)). Notably, this bound again incurs a penalty of $\mathcal{W}(\mathcal{D}_X, \tilde{\mathcal{D}}_X)$.

*Remark* C.8. Note that Eq. (45) suggests a general approach to select the distribution $\tilde{\mathcal{D}}_X$ that generates the training samples $\tilde{\mathbb{S}}_n$. In order to ensure small generalization gap, we need to focus on two terms depending on $\tilde{\mathcal{D}}_X$: (1) the variance term $\mathrm{Var}_{\tilde{\mathcal{D}}_X}(\ell_{\mathrm{d}, f}^h(\tilde{x}))$; and (2) the divergence term $\mathcal{W}(\mathcal{D}_X, \tilde{\mathcal{D}}_X)$. We note that finding a distribution that jointly minimizes both terms is a non-trivial task. That said, in our sampling approach in Eq. (5), we control for $\mathcal{W}(\mathcal{D}_X, \tilde{\mathcal{D}}_X)$ by operating in the latent space of a good quality teacher generative model and minimize variance by finding points with high loss values through gradient ascent, thereby striking a balance between the two objectives.

## D  TOOLBOX

This section presents necessary definitions and lemmas that we utilize to establish our theoretical results presented in Section 3 (and restated in Appendix C).

**Definition D.1** (Wasserstein-1 metric). Let $(\mathcal{X}, \rho)$ be a metric space. Given two probability distributions $\mathcal{D}_X^1$ and $\mathcal{D}_X^2$ over $\mathcal{X}$, Wasserstein-1 distance between $\mathcal{D}_X^1$ and $\mathcal{D}_X^2$ is defined as follows.

$$\mathcal{W}(\mathcal{D}_X^1, \mathcal{D}_X^2) := \inf_{\pi \in \Pi(\mathcal{D}_X^1, \mathcal{D}_X^2)} \mathbb{E}_{X, X' \sim \pi}\left[d(X, X')\right] = \inf_{\pi \in \Pi(\mathcal{D}_X^1, \mathcal{D}_X^2)} \int_{\mathcal{X} \times \mathcal{X}} \rho(X, X') \, d\pi(x, x'), \tag{46}$$

where $\Pi(\mathcal{D}_X^1, \mathcal{D}_X^2)$ denotes the set of all joint distributions over $\mathcal{X} \times \mathcal{X}$ that have $\mathcal{D}_X^1$ and $\mathcal{D}_X^2$ as their marginals.

**Lemma D.2** (Kantorovich-Rubinstein duality (Villani, 2008)). *Let* $\mathrm{Lip}_1(\rho)$ *denote the set of all 1-Lipschitz functions in the metric* $\rho$, *i.e., for any* $f \in \mathrm{Lip}_1(\rho)$,

$$|f(x) - f(x')| \leq \rho(x, x'), \ \forall \, x, x'. \tag{47}$$

*Then,*

$$\mathcal{W}(\mathcal{D}_X^1, \mathcal{D}_X^2) = \sup_{f \in \mathrm{Lip}_1(\rho)} \left( \mathbb{E}_{X \sim \mathcal{D}_X^1}[f(X)] - \mathbb{E}_{X' \sim \mathcal{D}_X^2}[f(X')] \right). \tag{48}$$

**Lemma D.3.** *Let* $\ell_{\mathrm{d}} : \mathbb{R}^K \times \mathbb{R}^K \to \mathbb{R}$ *be a loss function employed during the distillation. For a given teacher* $h : \mathcal{X} \to \mathbb{R}^K$ *and a function class* $\mathcal{F}$, *we assume the the induced function class*

$$\mathcal{G}_{\ell_{\mathrm{d}}, \mathcal{F}}^h = \{x \mapsto \ell_{\mathrm{d}}(f(x), h(x)) : f \in \mathcal{F}\} \tag{49}$$

*is contained in the class of L-Lipschitz functions with respect to a metric* $\rho$. *Then, for any two distributions* $\mathcal{D}_X^1$ *and* $\mathcal{D}_X^2$, *we have*

$$R_{\ell_{\mathrm{d}}, f}^h(\mathcal{D}_X^1) - R_{\ell_{\mathrm{d}}, f}^h(\mathcal{D}_X^2) \leq \mathcal{W}(\mathcal{D}_X^1, \mathcal{D}_X^2), \quad \forall \, f \in \mathcal{F}, \tag{50}$$

*where* $\mathcal{W}(\mathcal{D}_X^1, \mathcal{D}_X^2)$ *denotes the Wasserstein-1 metric between the two distribution* $\mathcal{D}_X^1$ *and* $\mathcal{D}_X^2$ *(cf. Definition D.1).*

*Proof.* Note that

$$
\begin{aligned}
R^h_{\ell_{\mathrm{d}},f}(\mathcal{D}^1_X) - R^h_{\ell_{\mathrm{d}},f}(\mathcal{D}^2_X) &= \mathbb{E}_{X\sim\mathcal{D}^1_X}\left[\ell_{\mathrm{d}}(f(X),h(X)\right] - \mathbb{E}_{X'\sim\mathcal{D}^2_X}\left[\ell_{\mathrm{d}}(f(X'),h(X')\right] \\
&\leq \sup_{g\in\mathcal{G}^h_{\ell_{\mathrm{d}},\mathcal{F}}}\left(\mathbb{E}_{X\sim\mathcal{D}^1_X}\left[g(X)\right] - \mathbb{E}_{X'\sim\mathcal{D}^1_X}\left[g(X')\right]\right) \\
&\overset{(i)}{=} L\cdot\sup_{g\in\mathcal{G}^h_{\ell_{\mathrm{d}},\mathcal{F}}}\left(\mathbb{E}_{X\sim\mathcal{D}^1_X}\left[\frac{g(X)}{L}\right] - \mathbb{E}_{X'\sim\mathcal{D}^1_X}\left[\frac{g(X')}{L}\right]\right) \\
&\overset{(ii)}{\leq} L\cdot\sup_{g\in\mathrm{Lip}_1(\rho)}\left(\mathbb{E}_{X\sim\mathcal{D}^1_X}\left[g(X)\right] - \mathbb{E}_{X'\sim\mathcal{D}^1_X}\left[g(X')\right]\right), \\
&\overset{(iv)}{=} \mathcal{W}(\mathcal{D}^1_X,\mathcal{D}^2_X).
\end{aligned}
\tag{51}
$$

where $(i)$ follow by dividing and multiply by $L$; $(ii)$ follows as, for any $g\in\mathcal{G}^h_{\ell_{\mathrm{d}},\mathcal{F}}$ is $\frac{g}{L}$ is 1-Lipschitz; and $(iii)$ follows from Lemma D.2. $\qquad\square$

**Lemma D.4.** *Let the distillation loss $\ell_{\mathrm{d}}$ satisfy Assumption C.1 with a bounded loss function $\ell:\mathbb{R}^K\times\mathcal{Y}\to\mathbb{R}$. Then, given a teacher $h:\mathcal{X}\to\mathbb{R}^K$ and a student model $f:\mathcal{X}\to\mathbb{R}^K$, we have*

$$
\left|R^h_{\ell_{\mathrm{d}},f}(\mathcal{D}_X) - R_{\ell,f}(\mathcal{D})\right| \leq \mathcal{O}\left(\sqrt{K}\cdot\mathbb{E}_{\mathcal{D}_X}\left[\|\mathcal{D}_{Y|X}-h(X)\|_2\right]\right),
\tag{52}
$$

*where $\mathcal{D}_{Y|X} = (\mathcal{D}_{Y|X}(1),\dots,\mathcal{D}_{Y|X}(K))$ is treated as a vector in $\mathbb{R}^K$.*

*Proof.* Note that

$$
\begin{aligned}
\left|R^h_{\ell_{\mathrm{d}},f}(\mathcal{D}_X) - R_{\ell,f}(\mathcal{D})\right| &= \left|\mathbb{E}_{\mathcal{D}_X}\left[\ell_{\mathrm{d}}(f(X),h(X))\right] - R_{\ell,f}(\mathcal{D})\right| \\
&= \left|\mathbb{E}_{\mathcal{D}_X}\left[\ell_{\mathrm{d}}(f(X),h(X))\right] - \mathbb{E}_{\mathcal{D}}\left[\ell(f(X),Y)\right]\right| \\
&= \left|\mathbb{E}_{\mathcal{D}_X}\left[\sum_{y\in[K]}h(X)_y\cdot\ell(f(x),y)\right] - \mathbb{E}_{\mathcal{D}_X}\left[\sum_{y\in[K]}\mathcal{D}_{Y|X}(y)\cdot\ell(f(X),y)\right]\right| \\
&= \left|\mathbb{E}_{\mathcal{D}_X}\left[\sum_{y\in[K]}\left(h(X)_y-\mathcal{D}_{Y|X}(y)\right)\cdot\ell(f(X),y)\right]\right| \\
&\overset{(i)}{\leq} \mathbb{E}_{\mathcal{D}_X}\left[\|\mathcal{D}_{Y|X}-h(X)\|_2\cdot\|\ell(f(X))\|_2\right],
\end{aligned}
\tag{53}
$$

where $(i)$ follow from the Cauchy-Schwarz inequality. Now the statement of Lemma D.4 follows from the assumption on the loss $\ell$ is bounded. $\qquad\square$

# E  ADDITIONAL EXPERIMENTS

## E.1  LONG-TAIL IMAGE CLASSIFICATION

Please see Table 4 for Places365-LT result. The relevant discussion is provided in Section 4.1. We also provided an expanded version of Table 1 (from the main text) in Table 5 with additional baselines.

# F  DETAILS TO REPRODUCE OUR EMPIRICAL RESULTS

Hereby we provide details to reproduce our experimental results.

## F.1  LONG-TAIL IMAGE CLASSIFICATION (SEC. 4.1)

**Dataset.** The full balanced version of 3 datasets (ImageNet [4], Place365 [5], SUN397 [6]) are available in tensflow-datasets (https://www.tensorflow.org/datasets/). Next to obtain the the

---

[4]`https://www.tensorflow.org/datasets/catalog/imagenet2012`
[5]`https://www.tensorflow.org/datasets/catalog/places365_small`
[6]`https://www.tensorflow.org/datasets/catalog/sun397`

| | Approach | Architecture | Balanced Accuracy | # parameters | FLOPs |
|---|---|---|---|---|---|
| Places365-LT | LWS (Kang et al., 2020) | ResNet-152 | 37.6 | 60 M | 11 B |
| | LDAM-DRS-RSG (Wang et al., 2021) | ResNet-152 | 39.3 | 60 M | 11 B |
| | OLTR (Liu et al., 2019b) | ResNet-152 | 35.9 | 60 M | 11 B |
| | DRAGON + Bal'Loss (Samuel et al., 2021) | ResNet-50 | 38.1 | 26 M | 4.1 B |
| | *Teacher (labeler) model* | EfficientNet-b3 | 42.1 | 12 M | 1.8 B |
| | One-hot | MobileNetV3-0.75 | 26.8 | 4.01 M | 156 M |
| | Distillation | MobileNetV3-0.75 | 33.0 | 4.01 M | 156 M |
| | TGT (random) | MobileNetV3-0.75 | 34.7 | 4.01 M | 156 M |
| | TGT (gradient-based) | MobileNetV3-0.75 | 35.0 | 4.01 M | 156 M |

Table 4: Performance of TGT on Places-LT (Liu et al., 2019c). The table shows the top-1 accuracy on the corresponding balanced eval sets for TGT and different long-tail baselines from the literature (taken from (Samuel et al., 2021)). We also state the number of model parameters and inference cost (in terms of FLOPs) for all the methods. Note that TGT leads to performance improvements over standard distillation. Note that, for Places-LT, TGT does not outperform stated baselines for the literature that rely on specialized loss functions and/or training procedures designed from the long-tail setting. One reason for this could be that the BigBiGAN does not generate very informative samples for Places-LT due to distribution mismatch. That said, as discussed in Section 4.1, one can combine the TGT framework with a long-tail specific loss functions as opposed to employing the standard cross-entropy loss function as a way to further improve its performance.

| | Approach | Architecture | Balanced Accuracy | # parameters | FLOPs |
|---|---|---|---|---|---|
| ImageNet K-LT | LDAM-DRW* (Cao et al., 2019) | ResNet-50 | 47.8 | 26 M | 4.1 B |
| | LWS (Kang et al., 2020) | ResNeXt-50 | 49.9 | 25 M | 4.2 B |
| | Logit adjustment loss* (Menon et al., 2021b) | ResNet-50 | 50.4 | 26 M | 4.1 B |
| | LDAM-DRS-RSG (Wang et al., 2021) | ResNeXt-50 | 51.8 | 25 M | 4.2 B |
| | DRAGON + Bal'Loss (Samuel et al., 2021) | ResNet-10 | 46.5 | 5.4 M | 819 M |
| | DRAGON + Bal'Loss (Samuel et al., 2021) | ResNet-50 | 53.5 | 26 M | 4.1 B |
| | *Teacher (labeler) model* | EfficientNet-b3 | 79.2 | 12 M | 1.8 B |
| | One-hot | MobileNetV3-0.75 | 35.5 | 4.01 M | 156 M |
| | Distillation | MobileNetV3-0.75 | 47.2 | 4.01 M | 156 M |
| | TGT (random) | MobileNetV3-0.75 | 53.2 | 4.01 M | 156 M |
| | TGT (gradient-based) | MobileNetV3-0.75 | 53.3 | 4.01 M | 156 M |
| SUN397-LT | LDAM-DRS-RSG (Wang et al., 2021) | ResNeXt-50 | 29.8 | 25 M | 4.2 B |
| | CAD-VAE (Schönfeld et al., 2019) | ResNet-101 | 32.8 | 42 M | 7.6 B |
| | LWS (Kang et al., 2020) | ResNeXt-50 | 33.9 | 25 M | 4.2 B |
| | DRAGON + Bal'Loss (Samuel et al., 2021) | ResNet-101 | 36.1 | 42 M | 7.6 B |
| | *Teacher (labeler) model* | EfficientNet-b3 | 65.3 | 12 M | 1.8 B |
| | One-hot | MobileNetV3-0.75 | 39.3 | 4.01 M | 156 M |
| | Distillation | MobileNetV3-0.75 | 42.2 | 4.01 M | 156 M |
| | TGT (random) | MobileNetV3-0.75 | 44.3 | 4.01 M | 156 M |
| | TGT (gradient-based) | MobileNetV3-0.75 | 44.7 | 4.01 M | 156 M |

Table 5: Performance of TGT and various baselines from the literature on ImageNet-LT and SUN-LT. Note that this table expands Table 1 (in the main text) as it includes additional baselines. Rows with * denote results taken from Menon et al. (2021b) and the rest were taken from Samuel et al. (2021). We report top-1 accuracy on balanced eval sets. We also state the number of model parameters and inference cost (in terms of FLOPs) for all the methods. Note that TGT leads to performance improvements over standard distillation on both datasets, particularly for ImageNet-LT where the teacher generator models the task distribution well. TGT also often outperforms stated baselines that rely on much larger and expensive models.

long-tail version of the datasets, we downloaded [7] image ids from repository of "Large-Scale Long-Tailed Recognition in an Open World (Liu et al., 2019b)" according to which we subsampled the full balanced dataset.

**Teacher fine-tuning.** For teacher labeler, we follow "Sharpness Aware Minimization' (Foret et al., 2020) codebase (available at `https://github.com/google-research/sam`) to fine-tune on the long-tail datasets. We start with pretrained EfficientNet-B3 model checkpoint available from

---

[7]`https://drive.google.com/drive/u/1/folders/1j7Nkfe6ZhzKFXePHdsseeeGI877Xu1yf`

official repository[8] and used default parameters from the codebase. We fine-tuned all 3 datasets (ImageNet-LT, SUN397-LT, Place365-LT) for 3 epochs.

We directly used teacher generator as BigBiGAN ResNet-50 checkpoint from the official repository `https://github.com/deepmind/deepmind-research/tree/master/bigbigan`. (We did not fine-tune it.)

**Student training.** We start from randomly initialized MobileNetV3-0.75 model. We employed SGD optimizer with cosine schedule (peak learning rate of 0.4 and decay down to 0). We also did a linear warm-up (from 0 to peak learning rate of 0.4) for first 5 epochs. The input image size are unfortunately different between EfficientNet-B3 model, BigBiGAN-ResNet50, and MobileNetV3-0.75 models. From original images in dataset, we use Tensorflow's bicubic resizing to obtain appropriate size image for each mode. We did a grid search over the perturbation parameters $\sigma$ and $\eta$ (c.f. Eq. (4) and Eq. (5)). All hyper-parameters and grid are listed in table below:

| Hyper-param | ImageNet-LT | Place365-LT | Sun397-LT |
|---|---|---|---|
| Num epochs | 90 | 30 | 30 |
| Optimizer | | SGD | |
| Schedule | | Cosine | |
| Warm-up epochs | | 5 | |
| Peak learning rate | | 0.4 | |
| Batch size | | 256 | |
| Teacher labeler image size | | $300 \times 300 \times 3$ | |
| Teacher generator image size | | $256 \times 256 \times 3$ | |
| Student image size | | $224 \times 224 \times 3$ | |
| Perturbation noise ($\sigma$) | | $\{0, 0.001, 0.01, 0.1\}$ | |
| Gradient exploration | | | |
| - Step size ($\eta$) | | $\{0, 0.001, 0.01, 0.1\}$ | |
| - Num steps | | 2 | |

Table 6: Hyper-parameters for long-tail image classification

### F.2 TGT IN LOW-DATA REGIME (SEC. 4.2)

**Dataset.** We used ImageNet [9] dataset from tensflow-datasets repository (`https://www.tensorflow.org/datasets/`). We used in-built sub-sampling functionality available in tensorflow (`https://www.tensorflow.org/datasets/splits`) to simulate the low-data regime.

**Teacher model.** For teacher labeler, we directly used trained EfficientNet-B3 model checkpoint available from "Sharpness Aware Minimization" repository[10] For teacher generator, we directly used trained BigBiGAN checkpoint from the official repository `https://github.com/deepmind/deepmind-research/tree/master/bigbigan`. (We did not fine-tune either of the models.)

**Student training.** We start from randomly initialized MobileNetV3-0.75 model. We employed SGD optimizer with cosine schedule (peak learning rate of 0.4 and decay down to 0). We also did a linear warm-up (from 0 to peak learning rate of 0.4) for first 5 epochs. The input image size are unfortunately different between EfficientNet-B3 model, BigBiGAN-ResNet50, and MobileNetV3-0.75 models. From original images in dataset, we use Tensorflow's bicubic resizing to obtain appropriate size image for each mode. Following standard practice in literature He et al. (2016); Jia et al. (2018), we train one-hot and standard distillation student models for 90 epochs (= 450k steps). We use 4x less steps for TGT than the simple distillation baseline, which amounts to 450k/4 = 112k steps.

---

[8] `https://storage.googleapis.com/gresearch/sam/efficientnet_checkpoints/noisystudent/efficientnet-b3/checkpoint.tar.gz`

[9] `https://www.tensorflow.org/datasets/catalog/imagenet2012`

[10] `https://storage.googleapis.com/gresearch/sam/efficientnet_checkpoints/noisystudent/efficientnet-b3/checkpoint.tar.gz`

### F.3 TEXT CLASSIFICATION (SEC. 4.3)

**Dataset.** We conduct text classification experiments on following datasets:

- Amazon-5 downloaded from `http://goo.gl/JyCnZq`
- IMDB from tensorflow-datasets `https://www.tensorflow.org/datasets/catalog/imdb_reviews`
- MNLI from from tensorflow-datasets `https://www.tensorflow.org/datasets/catalog/multi_nli`
- Yelp-5 downloaded from `http://goo.gl/JyCnZq`

**Optimizer.** For all training, we employed ADAM optimizer with linear decay schedule (peak learning rate of 3e-5 and decay to 0). We also did a linear warm-up at start. We used batch size of 128.

**Teacher fine-tuning.** For teacher labeler, we started from RoBERTa-Base (Liu et al., 2019a) pretrained checkpoint [11] from official FAIRSEQ repository `https://github.com/facebookresearch/fairseq`. We fine-tuned using default parameters, other than number of steps which are same as those listed in Table 7.

For teacher generator, we directly use a pre-trained BART-Base (Lewis et al., 2020) checkpoint [12] from official FAIRSEQ repository `https://github.com/facebookresearch/fairseq`. (We did not fine-tune it.)

**Student training.** We start from DistillBERT pretrained checkpoint downloaded from HuggingFace repository [13]. We perturb by adding Gaussian noise of $\sigma^2$ variance in between encoder-decoder as well as masking out $p$ fraction of input. Then we generate new examples by running a greedy decoding of BART teacher generator for sequence length of 512. For dual input classification task, like in MNLI, we generate the two inputs independently. We did a grid search over the perturbation parameters $\sigma$ and masking fraction $p$. All hyper-parameters and grid are listed in table below:

| Hyper-param | Amazon-5 | | IMDB | MNLI | Yelp-5 | |
|---|---|---|---|---|---|---|
| | 2.5k | 3M | | | 2.5k | 650k |
| Num steps | 5000 | 75000 | 20000 | 75000 | 5000 | 75000 |
| Warm-up steps | 1000 | 2000 | 500 | 2000 | 1000 | 2000 |
| Optimizer | | | Adam | | | |
| Schedule | | | Linear | | | |
| Peak learning rate | | | 3e-5 | | | |
| Batch size | | | 128 | | | |
| Max Sequence length | | | 512 | | | |
| Perturbation noise ($\sigma$) | | | {0, 0.01, 0.1} | | | |
| Masking fraction ($p$) | | | {0, 0.1, 0.2} | | | |

Table 7: Hyper-parameters for student training of text classification

### F.4 TEXT RETRIEVAL (SEC. 4.4)

**Dataset.** From official "Dense Passage Retrieval" repository at `https://github.com/facebookresearch/DPR`, we download passage corpus [14]. Further, from the same repository, we download a pre-processed version of natural questions open dataset (Lee et al., 2019) which has been aligned to passage corpus [15]. Finally, we download a pre-processed version of

---

[11] `https://dl.fbaipublicfiles.com/fairseq/models/roberta.base.tar.gz`

[12] `https://dl.fbaipublicfiles.com/fairseq/models/bart.base.tar.gz`

[13] https://huggingface.co/distilroberta-base/tree/main

[14] `https://dl.fbaipublicfiles.com/dpr/wikipedia_split/psgs_w100.tsv.gz`

[15] `https://dl.fbaipublicfiles.com/dpr/data/retriever/biencoder-nq-train.json.gz,https://dl.fbaipublicfiles.com/dpr/data/retriever/biencoder-nq-dev.json.gz`

Probably Asked Questions (PAQ) dataset (Lewis et al., 2021) dataset from official repository of "Domain-matched Pre-training Tasks for Dense Retrieval" available at `https://github.com/facebookresearch/dpr-scale` which has been aligned to the same passage corpus [16]

**Optimizer.** For all text retrieval model training, we employed ADAM optimizer with linear decay schedule (peak learning rate of 1e-5 and decay to 1e-7). We also did a linear warm-up (from 0 to peak learning rate of 1e-5) for 1K steps. We used batch size of 128.

**Teacher fine-tuning.** For teacher labeler dual encoder (a question encoder and a passage encoder), we utilized RoBERTa-Base (Liu et al., 2019a) pretrained checkpoint [17] from official FAIRSEQ repository `https://github.com/facebookresearch/fairseq`. We then conducted first round of fine-training for 300k iterations with passage-aligned PAQ dataset. We used same configuration as Oğuz et al. (2021) except Oğuz et al. trained with PAQ longer. After the pretraining, the teacher is fine-tuned on NQ-open (Kwiatkowski et al., 2019) downloaded with 40K steps. Similar to Karpukhin et al. (2020); Oğuz et al. (2021), the teacher is trained with within-batch negatives and the softmax-based cross-entropy loss.

For teacher generator, we directly use a pre-trained BART-Base (Lewis et al., 2020) checkpoint [18] from official FAIRSEQ repository `https://github.com/facebookresearch/fairseq`. (We did not fine-tune it.)

*This same teacher labeler and generator is used for all student training except for the direct training (one-hot).*

**Student training.** We start from DistillBERT pretrained checkpoint downloaded from HuggingFace repository [19]. All students are trained with 40K steps. The teacher labeler will label all-pair within the batch and will label additional 2 passages per each question-passage pair for the uniform negative sampling baseline and TGT. We employed a off-the-shelf BART-base model as our generator (Lewis et al., 2020) and isotropic perturbation was added by random Gaussian noise of scale $\sigma = 0.1$ combined with $p = 0.2$ for masking the original passage.

# G    QUALITATIVE EXAMPLES OF GENERATED EXAMPLES

## G.1    IMAGE CLASSIFICATION

We show some representative examples of generated images using TGT-random as well as TGT-gradient based from the experiment on ImageNet classification in Table 8.

## G.2    TEXT CLASSIFICATION

We show some representative examples of generated text using TGT from the experiment on MNLI classification in Table 9.

---

[16] `https://dl.fbaipublicfiles.com/dpr_scale/paq/PAQ.dpr.train.neg1.jsonl.zip`

[17] `https://dl.fbaipublicfiles.com/fairseq/models/roberta.base.tar.gz`

[18] `https://dl.fbaipublicfiles.com/fairseq/models/bart.base.tar.gz`

[19] https://huggingface.co/distilroberta-base/tree/main

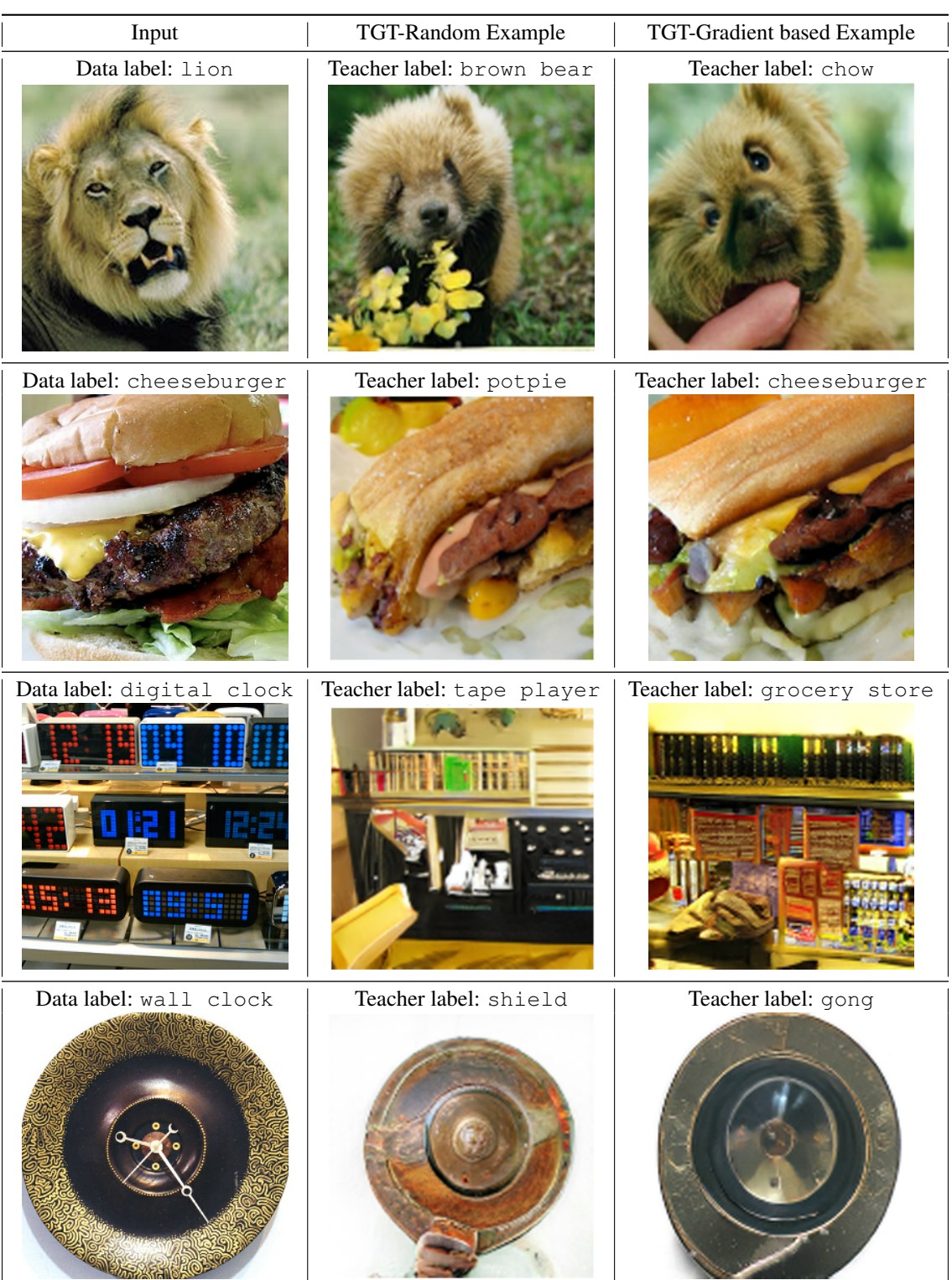

Table 8: Image examples

| Input | TGT Example |
|---|---|
| Data label: `Contradicts`

The house was bought with the royalties she earned from her first book, The Tales of Peter Rabbit. [SEP] The house was bought with the money she inherited from her grandfather. | Teacher label: `Neutral`

The book was published in the United States in 1987 with the royalties she received from her first book, The Tales of Peter Rabbit. [SEP] The house was bought with the money she inherited from her grandfather. |
| Data label: `Entail`

Leather goods are no longer a bargain in Spain, though very good quality products may still be priced lower than at home. [SEP] Leather goods are still very cheap in Spain. | Teacher label: `Entail`

Leather and leather goods are no longer a bargain in Spain, though very good quality products may still be priced lower than at home and abroad. [SEP] Leather goods are still very cheap at Spain. |
| Data label: `Entail`

Then I got up as softly as I could, and felt in the dark along the left-hand wall. [SEP] The wall was wet. | Teacher label: `Neutral`

Then I got up as softly as I could, and walked the way I felt in the dark along the left [SEP] The wall was wet. |
| Data label: `Entails`

But then this very particular island is hardly in danger of being invaded except, of course, by tourism. [SEP] This island is least likely to be invaded by tourism. | Teacher label: `Entail`

But then this very particular island is not in danger of being invaded except, of course, by tourism. [SEP] The island is likely to be invaded by tourism. |
| Data label: `Contradicts`

All you need to do is just wander off the beaten path, beyond the bustling tourist zone. [SEP] There is no point going off the beaten path, there is nothing there. | Teacher label: `Neutral`

All you need to do is just wander off the beaten path, and youĺl be in the bustling tourist zone of the city. [SEP] There is no point going off the beaten path, there is nothing there. |
| Data label: `Entails`

The silt of the River Maeander has also stranded the once-mighty city of Miletus. [SEP] The River Maeander has been depositing silt near Miletus for nearly two millennia. | Teacher label: `Neutral`

The silt of the River Mae has also stranded the once-mighty city of Miletus. [SEP] The River Maeander has been depositing silt near Miletus for more than two decades. |
| Data label: `Entails`

It was hardly the most enlightened of times, not with the conflict in Indochina rapidly becoming Americaś costliest and most divisive war. [SEP] The war in Indochina has cost America 100 billion dollars so far. | Teacher label: `Entails`

It was hardly the most enlightened of times, not with the war in Indochina becoming Americaś costliest and most divisive war. [SEP] The war in Indochina has cost America 100 billion dollars so far. |

Table 9: Text examples

