# OpenReview forum: "Teacher Guided Training: An Efficient Framework for Knowledge Transfer"
_ICLR.cc/2023/Conference — ICLR 2023 poster_

### Official Review · Reviewer_PrCv · 2022-10-24

**Confidence:** 3
**Correctness:** 3
**Technical Novelty And Significance:** 3
**Empirical Novelty And Significance:** 3
**Recommendation:** 6

**Clarity, Quality, Novelty And Reproducibility:**

Clarity: Good. This paper is well-written. It is enjoyable to read.

Quality: Good. Experiments are comprehensive and show the effectiveness of TGT.

Novelty: A little limited. Knowledge distillation is an existing approach.


**Strength And Weaknesses:**

Strengths:
+ This paper is well-written. It is enjoyable to read.
+ The proposed framework is simple and effective. The overall idea is neat.
+ Experiments are comprehensive and show the effectiveness of TGT.

Weaknesses:
+ The contributions are a little limited. Knowledge distillation is an existing approach.
+ More insightful analyses should be provided. How the knowledge is transferred in TGT?
+ Is the conclusion in 4.2 still hold true in NLP dataset?

**Summary Of The Paper:**

This paper targets solving knowledge transfer problems. Specifically, the authors distill large models to compact models for efficient machine learning. Teacher-guided training (TGT) framework is proposed for training high-quality compact models which learn knowledge from pretrained generative models. Experiments show the effectiveness of TGT.

**Summary Of The Review:**

Strengths:
+ This paper is well-written. It is enjoyable to read.
+ The proposed framework is simple and effective. The overall idea is neat.
+ Experiments are comprehensive and show the effectiveness of TGT.

Weaknesses:
+ The contributions are a little limited. Knowledge distillation is an existing approach.
+ More insightful analyses should be provided. How the knowledge is transferred in TGT?
+ Is the conclusion in 4.2 still hold true in NLP dataset?

---

> ### Author Response · Authors · 2022-11-13
> **Response to Reviewer PrCv**
>
> We appreciate the review and thank the reviewer for recognizing the value of the proposed framework and our effort in experiments.
>
> > The contributions are a little limited. Knowledge distillation is an existing approach.
>
> We wanted to highlight that our contribution is not proposing Knowledge Distillation. As in our extensive list of related works, Knowledge Distillation is a popular field of study with various techniques. Our contribution is proposing a novel methodology in the field of knowledge distillation, particularly a more efficient transfer of knowledge from large models to a compact model.  TGT relies on the fact that the teacher's internal representation of data often lies in a much smaller dimensional manifold than the input dimension. Furthermore, we can use the teacher to help guide training by identifying the directions where the student’s current decision boundary starts to diverge from that of the teacher, e.g., via backpropagating through the teacher to identify regions of disagreement. Thus, TGT adaptively changes the distribution of distillation examples, yielding higher performing student models with fewer training examples.
>
> Furthermore, we provide theoretical analysis for leveraging latent representations of the reach. Our analysis highlights the benefits of the TGT framework. We are not aware of any other work providing such generalization bounds.
>
> > More insightful analyses should be provided. How the knowledge is transferred in TGT?
>
> At a high-level, our proposal, TGT, leverages the knowledge of the teacher in two ways. First, TGT uses the teacher’s predictions and this is the same as the standard distillation. Second, TGT leverages the teacher’s knowledge of the data and actively finds important examples to be labeled. Also this is better than finding examples directly in the input space because the teacher's internal representation of data often lies in a much smaller dimensional manifold than the original data dimension. Because of this property, TGT can transfer the knowledge much better than the other methodologies. Please look at Appendix G for some illustrative examples of finding new data. Furthermore in Appendix H we added some visual high-level insights of our TGT method.
>
> > Is the conclusion in 4.2 still hold true in NLP dataset?
>
> Yes the conclusion in 4.2 holds for NLP dataset. Please see Section 4.3 and 4.4. Particularly in Section 4.3, we show results on sub-sampled version of the dataset and in some settings we were even able to completely remove the need of using a pre-trained student model, which demonstrates that the model does require a substantially smaller number of examples to achieve similar accuracy.

---

### Official Review · Reviewer_Qnjd · 2022-10-25

**Confidence:** 4
**Clarity, Quality, Novelty And Reproducibility:** The paper is well-written, and the pr…
**Correctness:** 3
**Technical Novelty And Significance:** 3
**Empirical Novelty And Significance:** 3
**Recommendation:** 6

**Strength And Weaknesses:**

Strength:

1. I think this paper will be valuable to the community. For the large pretrained models, even distilling to a lighter proxy can be unaffordable. Cleverly utilizing the teacher model and distilling the student efficiently is a very important problem. I like the theoretical analysis showing the benefit of utilizing TGT on generalization error bounds. These are good contributions.

2. The paper appropriately cites related works and discusses their relationship to the teacher-guided training method.

3. The paper is well-motivated and clearly written. Experiments are properly conducted to validate the effectiveness.

Weakness:

1. A minor disadvantage of TGT is that it relies upon an encoder-decoder style generative model as a teacher, which limits its practicability as most of the models trained in the industry are deterministic and do not have an encoder-decoder structure.

2. I am not sure if all three terms in Equation (2) are utilized in the experiments. If so, will the first supervised learning term dominate and nullify the better generalization applying TGT?

**Summary Of The Paper:**

The authors propose a general teacher-guided training (TGT) framework for efficient model distillation. TGT explicitly leverages the low dimensional representations extracted by the pretrained teacher generative models, which is then theoretically shown to be able to improve the generalization of the student model. TGT does not need to go through a large volume of data, thus and can better work in limited data regime or long-tail settings.

**Summary Of The Review:**

None

---

> ### Author Response · Authors · 2022-11-13
> **Response to Reviewer Qnjd**
>
> Thank you for reviewing our submission. We are glad that the reviewer recognized the novelty of our proposed TGT framework which is accompanied by theoretical justifications and has been validated to be effective in experiments. Here’s some clarifications on the weakness points that you made.
>
> > models trained in the industry are deterministic ...
>
> We would like to clarify a few points. First of all the beauty of the TGT framework is that the generator teacher can be different from the task-specific label teacher. One can use any publicly available off-the-shelf encoder-decoder model as the generator teacher (possibly fine-tuning it to their downstream task domain if needed). The labeling teacher could just remain intact, i.e. whatever model you are previously using. In some of our experiments reported in the paper also we have done this. Furthermore, we don’t need any stochastic teachers. Deterministic teacher models (labeller as well as generator) like T5, BART, GPT, etc. work. Note all of these models are from industry.
>
> > if all three terms in Equation (2) are utilized ...
>
> Regarding the point that you made for Equation (2), we respectfully disagree. In fact we have provided results based on each term in Table 1, 2, and 3. In particular, (1) the one-hot row corresponds to the first term, (2) the distillation row corresponds to the second term, (3) TGT rows correspond to using everything along with the third term with possibly different explorations. In addition we now provide additional experiments to augment the table on combining first and second terms for image datasets:
>
> | Approach | ImageNet-LT | SUN397-LT | Place365-LT |
> | --- | --- | --- | --- |
> | One-hot (first term)  | 35.5 | 39.3 | 26.8 |
> | Distillation (second term) | 47.2 | 42.2 | 33.0 |
> | One-hot + Distillation (first + second term) | 46.9 | 42.1 | 32.6 |
> | TGT(gradient-based) (all terms) | 53.3 | 44.7 | 35.0 |
>
> So clearly supervised learning term (first term) **does not** dominate and TGT **does** bring better generalization.
>
> Moreover, in literature often the first term (supervised learning term) is often dropped when we have good teacher labeler (cf. https://arxiv.org/pdf/1503.02531.pdf, https://arxiv.org/pdf/2005.10419.pdf)

---

### Official Review · Reviewer_hz78 · 2022-10-26

**Confidence:** 4
**Correctness:** 3
**Technical Novelty And Significance:** 2
**Empirical Novelty And Significance:** 3
**Recommendation:** 6

**Clarity, Quality, Novelty And Reproducibility:**

The presentation of this paper is of high quality. The theoretical analysis is insightful. However, I have some concerns on the novelty and reproducibility of this paper.

**Strength And Weaknesses:**

Strength:
- The problem is important. The proposed TGT framework is well-motivated.
- The experiments are extensive.
- A theoretical analysis is provided.

Weaknesses:
- It may be difficult to say that the proposed method is 'novel'. TGT is based on a standard knowledge distillation framework. The idea of 'minimizing the disagreement' is not new (e.g., [*1]). In general, I'm not surprised that this idea can improve the performance.
- From Table 1, one can see that TGT (gradient-based) only marginally outperforms the random baseline, i.e., TGT (random). In addition, the random baseline is not provided in other experiments. Therefore, I'm not fully convinced by the effectiveness of the proposed 'Gradient-based exploration' mechanism, which is an important contribution of this paper.
- In Table 1, the teacher model is pre-trained on a much larger dataset (i.e., JFT-300M) than the baselines. The comparison may be unfair. In addition, JFT-300M is not publicly available, which may hurt the reproducibility of this paper.

## Post-rebuttal
The responses from the authors have addressed most of my concerns (e.g., novelty and reproducibility). I have updated my score. However, I still think that adding more results of TGT (random) is important.


[*1] Miyato, T., Maeda, S. I., Koyama, M., & Ishii, S. (2018). Virtual adversarial training: a regularization method for supervised and semi-supervised learning. IEEE transactions on pattern analysis and machine intelligence, 41(8), 1979-1993.

**Summary Of The Paper:**

This paper studies an important problem, i.e., given a pre-trained large teacher model, how to transfer its knowledge to a compact low-compute smaller model. The proposed method is developed on top of the standard knowledge distillation approach. The authors assume that there is an available encoder and decoder, such that any training sample can be encoded into the latent space. Then one can find the maximally disagree point in the data manifold by exploring the latent space of the encoder, and train the student network by minimizing this disagreement. Extensive empirical results on both image and text data are provided.

**Summary Of The Review:**

Personally, I think that this paper is around the borderline. Currently, I'm slightly leaning reject. My rating may change after seeing the comments from other reviewers.

---

> ### Author Response · Authors · 2022-11-13
> **Response to Reviewer hz78**
>
> Thank you for taking the time to review our submission. We are happy to learn that the reviewer found our proposed method well motivated and recognized extensive empirical as well as theoretical studies.
>
> > The idea of 'minimizing the disagreement' ...
>
> Minimizing disagreement is the basis of most training objectives, even the most basic and common cross-entropy loss function, which minimizes the disagreement (in particular KL divergence) between true data distribution and model distribution. We are claiming the novelty lies in exploring so as to minimize the disagreement between the teacher and the student. We are not aware of any other work doing this to the best of our knowledge.
>
> The paper [*1] pointed out by the reviewer is doing adversarial training which is for smoothing the model distribution. It does this by finding the worst possible perturbation given the maximum range ($\epsilon$). Due to its adversarial nature, this method can make only a very small $\epsilon$ range and so can only do very small $\epsilon$-perturbation. Hence it does not provide diverse inputs whereas we don't have any such restrictions and we can traverse much further.
>
> > effectiveness of the proposed ...
>
> From the theorems, we see gains of TGT come from two places: 1) reduced dimension due to its use of latent space 2) better exploration for variance reduction. For the latter we proposed two strategies: a) a zero-order approach with random perturbation, “TGT (random)”, and b) a gradient-based first-order approach, “TGT (gradient-based)”. Thus both TGT(random) and TGT(gradient-based) are our key contributions.
>
> We find that on more long tail data (severely unbalanced) the gradient based approach helps more, however, most academic dataset is well balanced in which most of our win comes from utilizing the latent space, the form of search being of secondary importance. In Table 1, we quantify one such case on the SUN397-LT dataset where TGT (gradient-based) leads to improvement over TGT (random). In this dataset, we have an extremely long tail class distribution, i.e. 33 classes (~10%) have less than 5 images per class compared to ImageNet1K-LT where all classes have at least 5 images. This is because when data is extremely sparse, the exploration needs to be more dependent on the teacher's learnt data manifold and the gradient based method is more beneficial.
>
> Running all combinations is unfortunately not feasible due to the limited resources and time.
> We believe that the general study of identifying settings where first order search is better over zero-order search in general is beyond the scope of our paper and would form an interesting avenue for future research.
>
> > reproducibility of this paper
>
> We deeply care about reproducibility as well. We only used publicly available checkpoints from https://github.com/tensorflow/tpu/tree/master/models/official/efficientnet  and for the current submission we didn't use the JFT-300M dataset at all. The beauty of proposed TGT framework is that we can just use off-the-shelf teacher models, which might be trained by someone on their proprietary data, but we only need a small amount of end-task data (like ImageNet or SUN-397 in case of this paper) to transfer from teacher to student. This is even true for NLP domain where on many datasets we could match performance with students that have no pretraining at all.

---

### Official Review · Reviewer_u9os · 2022-10-27

**Confidence:** 5
**Correctness:** 4
**Technical Novelty And Significance:** 4
**Empirical Novelty And Significance:** 4
**Recommendation:** 8

**Clarity, Quality, Novelty And Reproducibility:**

Clarity

The paper is well written and easy to understand.

Reproducibility

The method is well explained and can certainly be reproduced.

Quality and Novelty

This paper represents a noteworthy advance over the state of the art. Good paper.

**Details Of Ethics Concerns:**

The authors present a new technique for compaction of large models. Large models are known to have various biases. Presumably the proposed compact version will also have those biases. The authors do not have an ethics section so it is not clear that they have taken ethical concerns into account. Perhaps it is just an inadvertent oversight that could be rectified easily with an added ethics section.

**Strength And Weaknesses:**

Strengths:

1. Innovative teacher-student distillation approach that is well motivated and explained.
2. Sound and insightful mathematical framework that enables application to both discrete and continuous domains.
3. Consistent and significant improvement over the state of the art in both model size reduction and accuracy.

Weaknesses
1. No significant weaknesses.

**Summary Of The Paper:**

The paper proposes a novel technique for producing compact versions of large models using a new approach to distillation named Teacher Guided Training. The authors develop a sound mathematical framework based on maximizing the match between the teacher and student model distributions.  The authors establish some useful performance bounds. The main insight of this work is in operating in the latent space which allows handling both discrete and continuous domains seamlessly. The technique also significantly improves upon the state of the art on a variety of test domains both in the reduction of parameters and the accuracy.

**Summary Of The Review:**

The paper nicely motivates the main techniques proposed. The literature survey is thorough and insightful. The proposed technique is backed up by a sound mathematical framework that is systematically established and well explained. The technique is shown to work well on a variety of domains and yields noteworthy improvement over the state of the art. It also advances the state of the art in the architecture and mathematics of its approach to the problem.

---

> ### Author Response · Authors · 2022-11-13
> **Response to Reviewer u9os**
>
> We thank the reviewer for the thorough review. We are glad that the reviewer found the work to be “sound and insightful mathematical framework” and “represents a noteworthy advance over the state of the art”.
>
> Regarding the ethics comments, we included an ethics section in the main text of the revised version.

---

### Decision · Program_Chairs · 2023-01-20

**Decision:**

Accept: poster

**Justification For Why Not Higher Score:**

The methodological novelty is a little bit limited.

**Justification For Why Not Lower Score:**

The method is simple and effective with some theoretical analysis.

**Metareview: Summary, Strengths And Weaknesses:**

In this paper, the authors proposed a simple way to improve the performance of knowledge distillation-based transfer learning methods with some theoretical analysis.

Though the novelty in terms of methodology is a concern, all reviewers agreed that the proposed approach is simple and effective, and the experimental results are promising. If all the concerns including adding additional experiments requested by the reviewers can be addressed in a revision, this work would become stronger.

**Note From Pc:**

if the above contains the word "oral" or "spotlight" please see: "oral" presentation means -> notable-top-5% and "spotlight" means -> notable-top-25%. As stated in our emails, we are disassociating presentation type from AC recommendations

**Summary Of Ac-Reviewer Meeting:**

When I sent a borderline papers list to SAC, I consider this as a borderline paper. However, after reading the authors' rebuttal, all reviewers have a consistent opinion: the proposed method is simple and effective though, in terms of methodology, it is not very novel. Therefore, I do not consider it a borderline paper anymore.